



# The Acceleration of Dissolved Cobalt's Ecological Stoichiometry due to Biological Uptake, Remineralization, and Scavenging in the Atlantic Ocean

Mak A. Saito[1*], Abigail E. Noble[1,2], Nicholas Hawco[1], Benjamin S. Twining[5],
Daniel C. Ohnemus[3], Seth G. John[3], Phoebe Lam[1,4], Tim M. Conway[6],
Rod Johnson[7], Dawn Moran[1], Matthew McIlvin[1]

[1]*Stanley W. Watson Biogeochemistry Laboratory, Marine Chemistry and Geochemistry Department, Woods Hole Oceanographic Institution, Woods Hole, MA USA 02543*
[2] *Gradient Corporation, 20 University Road, Cambridge, MA 02138*
[3] *University of Southern California, Department of Earth Sciences, Los Angeles, CA USA 90089.*
[4] *University of California Santa Cruz, Santa Cruz, CA USA 95064*
[5] *Bigelow Laboratory for Ocean Sciences, East Boothbay, ME USA 04544*
[6] *College of Marine Sciences, University of South Florida, St. Petersburg FL USA 33701*
[7] *Bermuda Institute of Ocean Sciences, St. Georges, Bermuda GE 01*

*corresponding author
November 23, 2016

*Submitted to Biogeosciences*



**Abstract**
Cobalt has the smallest oceanic inventory of all known inorganic micronutrients, and hence is particularly
vulnerable to influence by internal oceanic processes including euphotic zone uptake, remineralization,
and scavenging. Due to its small oceanic inventory, cobalt provides a unique case study for considering
the stoichiometric coupling between dissolved and particulate phases in the context of Redfield theory.
The ecological stoichiometry of total dissolved cobalt (dCo) was examined using data from a U.S. North
Atlantic GEOTRACES transect and from a zonal South Atlantic GEOTRACES-compliant transect
(GA03/3_e and GAc01), by Redfieldian analysis of its statistical relationships with the macronutrient
phosphate. Trends in the dissolved cobalt to phosphate (dCo:P) stoichiometric relationships were evident
in the basin scale vertical structure of cobalt, with positive dCo:P slopes in the euphotic zone and negative
slopes found in the ocean interior and in coastal environments. The euphotic positive slopes were often
found to accelerate towards the surface and this was interpreted as due to the combined influence of
depleted phosphate, phosphorus sparing mechanisms, increased alkaline phosphatase metalloenzyme
production (a zinc or perhaps cobalt enzyme), and biochemical substitution of Co for depleted Zn.
Consistent with this, dissolved Zn (dZn) was found to be drawn down to only twofold more than dCo,
despite being more than 18-fold more abundant in the ocean interior. Particulate cobalt concentrations
increased in abundance from the base of the euphotic zone to become ~10% of the overall cobalt
inventory in the upper euphotic zone with high stoichiometric values of ~400 $\mu$mol Co mol$^{-1}$ P.
Metaproteomic results from the Bermuda Atlantic Time-series Study (BATS) station found
cyanobacterial isoforms of the alkaline phosphatase enzyme to be prevalent in the upper water column, as
well as a sulfolipid biosynthesis protein indicative of P sparing. The negative dCo:P relationships in the
ocean interior became increasingly vertical with depth, and were consistent with the sum of scavenging
and remineralization processes (as shown by their dCo:P vector sums). Attenuation of the
remineralization with depth resulted in the increasingly vertical dCo:P relationships. Analysis of
particulate Co with particulate Mn and particulate phosphate also showed positive linear relationships
below the euphotic zone, consistent with the presence and increased relative influence of Mn oxide
particles involved in scavenging. Visualization of dCo:P slopes across an ocean section revealed hotspots
of scavenging and remineralization, such as at the hydrothermal vents and below the OMZ region,
respectively, while that of an estimate of Co* illustrated stoichiometrically depleted values in the
mesopelagic and deep ocean due to scavenging. This study provides insights into the coupling between
the dissolved and particulate phase that ultimately create Redfield stoichiometric ratios, demonstrating
that the coupling is not an instantaneous process and is influenced by the element inventory and rate of
exchange between phases. Cobalt's small water column inventory and the influence of external factors on
its biotic stoichiometry can erode its limited inertia and result in an acceleration of the dissolved
stoichiometry towards that of the particulate phase in the upper euphotic zone. As human use of cobalt
grows exponentially with widespread adoption of lithium ion batteries, there is a potential to alter this
dynamic biogeochemical cycling and ecology of cobalt in the oceanic euphotic zone.





## 1. Introduction

The biogeochemical cycle of cobalt is one of the more complex among trace metals present in the ocean. Complexities affecting cobalt include chemical processes such as redox transformations, complexation, low solubility and incorporation into mineral phases, and biological processes such as varying biochemical requirements and an early adoption of cobalt during Earth's biological and geochemical co-evolution. The nutritional importance of cobalt stems from its requirement in the biosynthesis of vitamin $B_{12}$ and subsequent requirements of the vitamin (Rodionov et al., 2003), as well as for its ability to substitute within a diatom carbonic anhydrase enzyme (Morel et al., 1994; Roberts et al., 1997). There are likely also other as yet undetermined biochemical functions of cobalt within both cyanobacteria, that have an absolute requirement for cobalt (Saito et al., 2002), and other eukaryotic phytoplankton that often show a physiological capacity for substitution of cobalt for zinc (Saito and Goepfert, 2008; Sunda and Huntsman, 1995). These cobalt micronutritional requirements have been proposed to be important in the ecology of phytoplankton, such as the marine cyanobacteria *Synechococcus,* as well as the coccolithophore *Emiliania huxleyi* (Ahlgren et al., 2014; Sunda and Huntsman, 1995).

The confluence of these chemical and biological processes results in cobalt having a complex elemental cycle that has been described as a "hybrid-type" of the nutrient and scavenged vertical profile categories (Bruland and Lohan, 2003; Noble et al., 2008). The utilization of cobalt as a nutrient by phytoplankton results in surface depletion as well as subsequent accumulation at intermediate depths through remineralization of sinking particulate material. In contrast, the scavenging process for cobalt is likely a one-way removal flux of dissolved cobalt from seawater that results in depletion at intermediate and deep ocean depths, and is thought to be driven by the co-oxidation of cobalt upon microbial oxidation and precipitation of manganese oxide around small neutrally buoyant bacteria (Cowen and Bruland, 1985; Lee and Tebo, 1994; Moffett and Ho, 1996; Tebo et al., 1984). The factors that affect cobalt cycling, including its small oceanic inventory, susceptibility to scavenging, utilization dependence on the availability of other micronutrients, and labile concentrations present in the water column, make it uniquely exposed to multiple processes that can have a dramatic effect on the vertical and sectional structure of its oceanic distributions.

The microbial ocean ecosystem's nutrient stoichiometry has been inferred by examination of both the elemental composition of dissolved and particulate phases. This approach was first pioneered by Alfred Redfield for dissolved and particulate nitrogen and phosphate (Redfield, 1958; Redfield et al., 1963). As a micronutrient with a very small oceanic inventory, cobalt provides a unique case study for considering the stoichiometric coupling between dissolved and particulate phases. When applied to cobalt, linear relationships between dissolved cobalt and soluble reactive phosphate (dCo:P) can be interpreted as a time-integrated signal of the extent of cobalt utilization by the resident phytoplankton community and their subsequent remineralization from the biological particulate phase. The aggregate slope of this correlation is often described as "ecological stoichiometry" for its inferred biological usage across a diversity of organisms present (Sterner and Elser, 2002). An emerging feature and mystery regarding cobalt relative to other macro (N and P) and micronutrients (Zn and Cd) is its unusually large range in stoichiometries spanning more than an order of magnitude from 29 μmol:mol in the Central North Pacific to 560 μmol:mol in the Equatorial Atlantic (Noble et al., 2012; Noble et al., 2008; Saito et al., 2010; Saito and Moffett, 2002). That high Equatorial Atlantic value has long appeared to be an outlier, and these putative high stoichiometries are a focus of this study. The biochemical basis for this stoichiometric variability remains unknown.





Within the ocean interior, cobalt scavenging is frequently described as a critically important
process for Co. But in actuality there has been little direct evidence provided to support this assertion.
Some of the available datasets include several process studies of Co and Mn radiotracer uptake into biotic
particles in the Sargasso Sea and coastal environments (Lee and Fisher, 1993; Moffett and Ho, 1996), a
study of Co and Mn precipitation in anoxic fjords (Tebo et al., 1984), and laboratory experiments
demonstrating that manganese oxidizing bacteria also oxidize Co under aerobic conditions (Lee and Tebo,
1994). These few studies, combined with early observations of the depletion of dissolved cobalt in
intermediate and deep waters (Knauer et al., 1982; Martin et al., 1989), and observations of a "cobalt
curl" in the plots of dissolved cobalt and phosphate space showing preferential removal of Co vs. P
(Noble et al., 2012; Saito et al., 2010), have contributed to this notion that scavenging is an important
process. A recent study put forth a contrary argument that scavenging is less important in the cycling of
dissolved cobalt, and that instead cobalt depletion in the ocean interior can be better explained by physical
and remineralization processes (Dulaquais et al., 2014b).
14         Improved analytical methods for dissolved Co and the resulting recent production of large
GEOTRACES datasets (Geotraces Group, 2015) provides new opportunities to explore the variability in
Co stoichiometry and the processes that create this range  (Baars and Croot, 2015; Bown et al., 2011;
Bown et al., 2012; Dulaquais et al., 2014a; Dulaquais et al., 2014b; Hawco et al., 2016; Noble et al.,
2012; Noble et al., 2008; Saito et al., 2010; Shelley et al., 2012). Here we examine and compare two
zonal full depth sections of total dissolved cobalt and labile cobalt from the North and South Atlantic
Ocean to examine the tug-of-war among competing processes affecting cobalt cycling in the oceanic
water column. Specifically, we have developed and employed a fine-scale Redfieldian statistical analysis
of the variation in the stoichiometry of cobalt relative to phosphorus across vertical and horizontal
dimensions to discern biogeochemical processes influencing dissolved cobalt in the Atlantic Ocean. This
manuscript is a companion to that of Noble et al., (submitted) describing the sources and distributions of
dissolved cobalt and its chemical speciation in the U.S. North Atlantic GEOTRACES GA03 section.
**2. Materials and Methods**
*Data Acquisition and Sources*
29         Total dissolved cobalt and cobalt speciation data utilized in this analysis were obtained from the
two legs of the U.S. North Atlantic GA03 and GA03_e Transect (2010 and 2011, also described as
USGT10 and USGT11, respectively) and the GEOTRACES-compliant CoFeMUG GAc01 transect
(Noble et al., 2012; Saito et al., 2013). All total dissolved cobalt and labile cobalt analyses (defined as
with and without UV irradiation, respectively) were conducted using cathodic stripping voltammetry
methods using dimethyl glyoxime as the added complexing agent, as described in the accompanying
manuscript (Noble et al., submitted) and in Noble et al., 2012 for GAc01. Both datasets utilized
GEOTRACES intercalibration standards to ensure the data are intercomparable.
*Statistical Analysis of Cobalt Stoichiometry*
39         Two-way linear regressions of dissolved cobalt and phosphate were conducted on both datasets
using a custom script written in MATLAB. This script called the external m-file lsqfitma.m, written by
Ed Peltzer for two-way linear regressions, which are the preferred approach to analysis of stoichiometry
because there is no assumption of dependence (parameter x on y or vice versa) between variables as exists
in standard least squares linear regressions (Glover et al., 2011). Regressions were performed using two
strategies: 1) on hand-selected depth ranges in the overall data by geographic region, and 2) on groups of





5-point adjacent datapoints moving downward point-by-point within each vertical profile. The latter analyses were conducted within Matlab scripts for the two-way linear regression analyses. Linear regression results used for figures were limited to those with correlation coefficient (r) values greater than $|0.7|$, with $r > 0.7$ correspond to positive slopes and $<-0.7$ for negative slopes. Matlab analysis scripts will be available at the Saito laboratory Github site for code (www.github.com/saitomics).

*Global Metaproteomic Analyses for Relevant Metalloenzymes*

Protein samples were collected from a Bermuda Atlantic Timeseries Study cruise (B313, April 2015) using *in-situ* McLane pumps deployed with mini-MULVFS filter heads (Bishop et al., 2012), and size-fractionated as previously described using 0.2, 3.0 and 51 micron filters (Saito et al., 2015; Saito et al., 2014). The 0.2 μm filter was extracted for total protein content of the microbial community using a SDS detergent and an adapted magnetic hydrophilic bead methodology (Hughes et al., 2014; Saito et al., 2014). The samples were analyzed by ultra-high-resolution mass spectrometry on a Thermo Fusion using TopN data dependent acquisition mode and dynamic exclusion of 30s. Two alkaline phosphatases were identified from the global metaproteome datasets using a custom metagenomics and genomic dataset as previously described that contains numerous cyanobacterial genomes (Saito et al., 2015; Saito et al., 2014). The full discussion of this metaproteome dataset is beyond the scope of this manuscript and will be described in a subsequent manuscript. The two alkaline phosphatase (PhoA) proteins identified corresponded to sequences from North Atlantic picocyanobacterial isolates *Prochlorococcus* NATL1 (gene 11501) and *Synechococcus* WH8102 (gene SYNW2391).

*Complementary Datasets: Particulate Metal and Phosphorus Datasets, Dissolved Zn and Cd, and Macronutrients*

Several datasets from the U.S. GA03/3_e 2010 and 2011 cruises (USGT10 and USGT11) North Atlantic Zonal Transect (NAZT) expedition were used to contribute to the interpretation of this study, including soluble reactive phosphate, particulate cobalt and phosphate, and dissolved zinc and cadmium. Particulate metal and phosphorus datasets were used for comparison to dissolved Co in this study. McLane pump small size fraction membrane filters (SSF, 0.8-51 μm, Pall Supor polyethersulfone membrane filters) were utilized for full ocean depth comparisons, and membrane filters from Go-Flo bottles (Pall Supor polyethersulfone membrane filters, 0.2 μm) were used for higher resolution upper water column comparisons. The data, as well as methods for collection, digestion, and analyses were previously described (Ohnemus and Lam, 2015; Twining et al., 2015). Methods for the dissolved zinc and cadmium measurements using isotope dilution on a multi-collector ICP-MS were described in Conway et al. (Conway et al., 2013), and the data were described in related North Atlantic manuscripts (Conway and John, 2015, 2014). Soluble reactive phosphorus was measured by nutrient auto-analyzer as described previously for the South Atlantic (Noble et al., 2012), and by the Scripps ODF facility for the North Atlantic.

## 3. Results and Discussion

### 3.1 Statistical Analysis of dCo:P Distributions in the North and South Atlantic Ocean

The study of nutrient stoichiometry has a long and important tradition in oceanography. For example, when nitrogen and cadmium concentrations have been compared relative to those of phosphorus, their linear relationships have been interpreted to imply an ecological use of those nutrients





throughout the microbial and phytoplankton community through their uptake and release from the
biological particulate phase (Boyle et al., 1976; Redfield et al., 1963; Sunda and Huntsman, 2000),
leading to the discovery of new metalloenzymes and the development of paleooceanographic proxies
(Boyle, 1988; Lane et al., 2005). The slopes of these relationships have been used to infer the ecological
stoichiometries of the biological processes that created them, based on the underlying idea that movement
of elements between the dissolved and biological particulate phases results in a "biochemical circulation"
of the oceans that is both distinct from and yet also overlaid upon the physical ocean circulation processes
(Redfield et al., 1963). Deviations in the stoichiometry of N:P inorganic chemical species (nitrate and
soluble reactive phosphate) in the environment have been described as evidence for non-Redfieldian
stoichiometry (Anderson and Pondaven, 2003; Arrigo et al., 1999). Similarly, "kinks" in the dissolved
Cd:phosphate relationships have been suggested to result from specific regional biogeochemical
processes such as iron-limitation, competition of ferrous iron at the cadmium transporter site, and
variations in Zn availability (Cullen et al., 2003; Lane et al., 2009; Sunda and Huntsman, 2000). For
cobalt, there is an emerging picture that its ecological stoichiometry is particularly complex, with larger
ecological stoichiometric variations compared to N and even trace elements such as Cd. Yet, as
previously suggested (Sunda and Huntsman, 1995), these Co:P signals likely provide important
information regarding the biological and chemical processes influencing dissolved cobalt distributions.
One of the outstanding questions regarding dissolved stoichiometries of metals and nutrients is the nature
and strength of their connection to the particulate phase.
In this study, total dissolved and labile datasets from the US North Atlantic GEOTRACES
Transect (GA03 and GA03_e) and the South Atlantic GEOTRACES-compliant CoFeMUG expedition
(GAc01; Fig. 1) were analyzed for their relationships relative to the macronutrient phosphate. These large
Atlantic datasets provided an opportunity to examine Co biogeochemical complexity and to test prior
hypotheses regarding micronutrition and element substitution. Simple two-way linear regressions were
employed in two manners for the examination of the ecological stoichiometry of dissolved cobalt and
phosphate ($dCo:PO_4^{3-}$; or dCo:P hereon; see methods for specific details). The first approach ("aggregate
regression" analysis hereon), resembled a standard Redfieldian analysis that determined dissolved cobalt
versus phosphate relationships within specific water parcels (subsets of the sections) with some selected
datapoints removed that were associated with proximity to coastal and hydrothermal regions. The
resulting dCo and phosphate distributions were visualized with respect to the origin of their water masses
(Fig. 2).
The second approach ("profile-based regression" approach hereon) involved an unbiased and
inclusive statistical analysis also using linear regressions, but now applied to a moving five-point depth
window on each individual vertical profile in order to capture fine-scale structural changes in Co
ecological stoichiometry (Fig. 3). Because the processes of remineralization of sinking biomass and
scavenging onto particles results in vertical transport through the water column, this profile-based
regression statistical analysis could be well-suited to detecting changes in stoichiometry of dissolved
metals relative to phosphate, and how those processes and their vertical signals are gradually integrated
across horizontally advected isopycnal surfaces. Notably, positive dCo:P (Figs. 3B and 3F) and negative
dCo:P (Figs. 3C and 3G) slopes were identified in both basins, implying trends where dissolved Co and
phosphate both increased, or dissolved Co decreased while phosphate increased, respectively. These
trends increased in magnitude towards the surface and deep respectively, as shown by visualizing
variation in dCo:P slope with depth (Figs. 3D and 3H), after filtering by correlation coefficients (r) values
above the $|0.7|$ threshold (equivalent to an $r^2 \geq 0.49$), and data below this threshold was excluded. A



simple schematic of the influence of phytoplankton uptake, remineralization, scavenging and dust input is
shown in Fig. 4(a) for comparison as well as actual measured vectors in Fig. 4(a) and (d) where each
dCo:P slope was represented as a vector in dCo and P space (vector lengths made uniform), and an
explanation for these observed negative slopes is presented later in the manuscript (Section 3.3). Note that
the depth associated with each linear regression result was assigned to the middle depth of the five depths
being analyzed, resulting in no regression results data at the upper and lower two depths of each profile.
Results of dCo:P linear regression for individual vertical profiles are shown in Figs. 5 (sign of slope by
color overlaid on dCo abundances) and 6 (magnitude of slope) to allow geospatial inspection and
comparisons. Interpretations of these trends and their variability in slopes measured by both aggregate
and profile regressions is discussed in the context of euphotic zone phytoplankton processes and
mesopelagic scavenging processes in the subsequent sections.
**3.2 Ecological Stoichiometry of Cobalt Across Transects of the Upper Atlantic Ocean**
**3.2.1 Distinct dCo:P Relationships in Mid-Euphotic to Upper Mesopelagic**
15          The aggregate regression analysis of the two zonal Atlantic Ocean sections found coherent large
scale linear relationships between dCo and phosphate in the mid-euphotic/upper mesopelagic. Distinct
dCo:P stoichiometries were identified ranging from 31-67 µmol mol$^{-1}$ ($r^2$ of 0.71 to 0.93) in this mid-
euphotic/upper mesopelagic, comparable to those observed in other geographic regions (Table 1). The
Atlantic data were grouped into five broad regions: the Eastern North Atlantic (USGT10-02 to USGT10-
06), the North Atlantic Subtropical Gyre (USGT11-10 to USGT11-23), the Mauritanian Upwelling
(USGT10-07 to USGT10-12 and USGT11-24), the South Atlantic Subtropical Gyre (CoFeMUG Stn 1-7),
and the Angola Gyre/Benguela Upwelling (CoFeMUG Stations 8-17; for individual regression plots see
Noble et al., submitted, their Fig 11.). There were several observations of note. First, the highest dCo:P
value, 67 µmol mol$^{-1}$ ($r^2 = 0.93$) observed between 135 m – 400 m depth, was observed in the North
Atlantic Subtropical Gyre where phosphate concentrations were extremely low (often < 0.01 µM within
the euphotic zone below 100 m). Second, the Eastern North Atlantic (USGT10 Stations 2-6) had a
particularly deep range where the dCo:P relationship was maintained (50 – 900 m, 41 µmol mol$^{-1}$, $r^2 =$
0.92), likely indicating the pronounced influence of remineralization processes. This feature spans three
water masses: North Atlantic Central Water, Atlantic Equatorial Water, and even Mediterranean Outflow
Water (the latter at 600-800 m for USGT10-2), and hence this dCo:P coherence could be due to a
regionally strong vertical influence from remineralization on the mesopelagic. Third, the region near and
within the productive Mauritanian Upwelling, with their contributions from Atlantic Equatorial Waters
(AEW) between 100-600 m depth (Jenkins et al., 2015), an oxygen minimum centered around 400 m (40-
110 µmol kg$^{-1}$ O$_2$), and atmospheric deposition from the Sahara Desert, had a dissolved stoichiometry of
48 µmol mol$^{-1}$ ($r^2 = 0.83$). Finally, linear regressions of aggregated data from the upper euphotic zone had
low coefficients of determination ($r^2$) and inspired the development of the profile-based methods
described below.
38          These dCo:P stoichiometries described above for the upper water column were likely controlled
by the aggregate influences of phytoplankton uptake and remineralization on dissolved cobalt throughout
this region, and their variability demonstrates how the influence of biological processes on cobalt
inventories is dynamic. For example, the tightest correlation and steepest slope were found in the North
Atlantic Subtropical Gyre (Table 1), excluding the profile-based analyses and the surface transect of the
Equatorial Atlantic (Saito and Moffett, 2002). These North Atlantic sites occurred where labile cobalt





concentrations were high and a strong correlation between labile cobalt and phosphate was also observed
(23 µmol mol$^{-1}$, $r^2 = 0.83$; Noble et al., submitted). The presence of a substantial labile cobalt pool was
consistent with the observed high cobalt usage, since labile cobalt is likely highly bioavailable relative to
complexed cobalt (Saito et al., 2002; Saito et al., 2005).
5        The South Atlantic zonal section had an ecological stoichiometry in the mid-euphotic/upper
mesopelagic zone (31 µmol mol$^{-1}$, $r^2 = 0.71$, 70-200 m) that was less than half that observed in the North
Atlantic Subtropical Gyre (67 µmol mol$^{-1}$; Table 1). The upper water column of the South Atlantic
Subtropical Gyre was characterized by strong complexation of cobalt and higher phosphate throughout
much of the region (Noble et al., submitted), both of which are consistent with a lower biological use of
Co, higher P use, and resultant lower dCo:P stoichiometry. While the Mauritanian Upwelling region also
showed evidence of a curve in the dCo:P relationship that shifts to a deeper ecological stoichiometry of
47 µmol mol$^{-1}$ (50-425 m, $r^2 = 0.83$), the Angola Gyre/Benguela Upwelling region did not demonstrate
changes in ecological stoichiometry with depth and the aggregate Co:PO$_4^{3-}$ ratio observed was 48 µmol
mol$^{-1}$ (0-400 m, $r^2 = 0.84$, Fig. 2). A potentially larger coastal sedimentary Co flux and entrainment in the
phytoplankton uptake and remineralization likely contributes to this higher ratio as a result of the lower
oxygen waters sampled in the Benguela Upwelling (Noble et al., 2012). Consistent with this sampling of
lower oxygen waters, the Benguela Upwelling region exhibits the highest total cobalt concentrations, yet
labile cobalt is low, resulting in lower Co:P stoichiometry and a weaker statistical relationship ($r^2 = 0.84$)
than in the North Atlantic Subtropical Gyre as described in Noble et al. (submitted).
**3.2.2 Evidence for Elevated Ecological Stoichiometries of Cobalt in the Upper Photic Zone**
22        The aggregate dCo:P relationships in the mid-photic zone and upper mesopelagic described above
excluded shallower and deeper depths that deviated from the linear relationships. Closer examination
revealed steeper coherent cobalt-phosphate relationships that do not appear to be random phenomena.
These deviations were observed above and below the selected mid-euphotic to upper mesopelagic depth
range studied above, and appeared to be caused by distinct biological and scavenging processes,
respectively. Specifically, these Atlantic datasets both showed the presence of an intriguing convex kink
and near-vertical cloud of cobalt data points near the x-axis as phosphate became increasingly depleted
(Fig. 2, blue arrows). While we have noticed these steep relationships previously (Saito et al., 2002;
Noble et al., 2012), dCo:P relationships at shallower depths were often statistically insignificant when
analyzed as part of a larger aggregated dataset as conducted in the previous section. The profile-based
regressions applied to the dCo:P dataset provided an effective alternative means to characterize these
steep features. Numerous positive relationships with steep slopes were observed in the dCo-phosphate
space (Fig. 3(b), (f)), particularly in close proximity to the surface in both the North and South Atlantic
basins as shown in dCo:P slopes of individual profiles (Figs. 5 and 6). These dCo:P slopes when
presented in vector form revealed a surprising extent of diversity with both positive and negative slopes
(Fig. 4(a), (c)). Moreover, the frequency of dCo:P slope stoichiometries in histogram form was highest
between 0-125 µmol:mol and decreased successively in the next three increasing bins of 125 µmol:mol in
both the North and South Atlantic. From this histogram it is apparent there was a broad tail of dCo:P
slope stoichiometries spanning from ~500 to -500 µmol:mol.
41        In an effort to understand the processes causing the variability in dCo:P slopes generated by the
profile-regression analysis, we focused on oligotrophic stations in the North Atlantic subtropical gyre
where these effects were particularly pronounced (USGT-2011 stations 14, 16, 18, and 20). Phosphate
abundances were particularly depleted in this region (Fig 7A), and the shallowest dCo:P stoichiometries





measured by the profile-based analysis were 320, 544, 491, and 197 µmol mol$^{-1}$ (Fig. 7(d)), centering
around depths of 101, 92, 76, and 76 m, Table 1). These values were roughly an order of magnitude
higher than the aggregate mid-euphotic upper mesopelagic zone values reported in the previous section
and observed in other studies (Table 1). Note that in the North Atlantic 40 m was the shallowest bottle
depth, and hence actual dissolved stoichiometries could have even higher values if depths above 40 m
were available for the 5-point regression analysis. Particulate Co and P (pCo and pP) concentrations
increased towards the surface with increasing photosynthetic activity (Fig. 7(b), (c)). The resulting
pCo:pP ratios were similar to those estimated from the dCo:dP profile-based slope analyses, approaching
~350-400 µmol mol$^{-1}$ (comparison of Fig. 7(d) and (e)). These observations were consistent with the high
dCo:P value of 560 µmol mol$^{-1}$ from a surface transect (5 m depth) across the Equatorial Atlantic, which
appeared to be an outlier due to being an order of magnitude higher than other literature values, as
mentioned above (Saito and Moffett, 2002). Interestingly, in some regions positive dCo:P slopes were
also observed below the 'cobalt-cline' such in the eastern basin of the North Atlantic (e.g., station 10-10,
Figs. 5-6), typically below 1000 m. This trend could reflect the influence of export and remineralization
of particles after they pass through the oxygen minimum zone where remineralization was slowed by
lower oxygen abundance, contributing an addition of dCo and phosphate overlaid on the small dCo
inventory.
**3.2.3 Observations and Explanations for the Acceleration of dCo:P Stoichiometries**

20          In addition to observations of high cobalt stoichiometries, the profile-based regression approach

also revealed a progression of successively increasing dCo:P slopes, or acceleration of dCo:P, towards the
ocean surface. This use of the term acceleration here is comparable to its use in physics where velocity is
the distance covered per unit time (first derivative), and acceleration is defined as increases in velocity
(second derivative). These velocity values were approximated simply by calculating the slope of tangents
by the profile-regression analysis, and the stoichiometric acceleration can then be observed in the
coherent increasing trends in these stoichiometric "velocities" with shallower depths. Similarly
decelerating stoichometries were observed moving into the ocean interior. The USGT11 stations 14, 16,
and 18 described above with depleted phosphate displayed this dCo:P acceleration towards the surface
(Fig. 7(d)), increasing by 1.8 µmol mol$^{-1}$ m$^{-1}$ between 288 m and 40 m, or 450 µmol mol$^{-1}$ within the
upper 250 m of station 18. Surveying the vertical profiles of dCo:P slopes (Fig. 6), a number of stations
included in this study display increasing dCo:P stoichiometry towards the water-atmosphere interface,
consistent with the acceleration of dCo:P data. This acceleration of the dissolved cobalt stoichiometry
towards the surface was greater in the North Atlantic relative to the South Atlantic (Fig 4C, F), where
dCo:P increased as with decreasing phosphate concentrations, reaching maximum values of
approximately 500 and 150 µmol mol$^{-1}$, respectively.

36          Together, these results imply the presence of a higher ecological stoichiometry of Co in the

oligotrophic gyre created by resident microbial and phytoplankton communities imprinting themselves on
the dissolved phase. There are three co-occurring phenomena that together likely explain these
observations of high and accelerating dCo:P stoichiometries in the upper oceanic photic zone: 1)
decreased biological use of phosphate due to its depletion in the upper water column and through sparing
mechanisms in phytoplankton, 2) substitution of Co for Zn within metalloenzymes as Zn depletion
occurred in the upper photic zone, and 3) enhanced Co and Zn metal nutritional needs due to biosynthesis
of the metalloenzyme alkaline phosphatases as a strategy for liberating dissolved organic phosphorus
upon phosphate scarcity. The first explanation would decrease the denominator, while the latter two




would act to increase the numerator of the dCo:P relationships.
With regards to the first process, the upper photic zone has highly depleted soluble reactive
phosphorus abundances (Fig. 7(a)). This is particularly true in the North Atlantic, which has the lowest
reported phosphate abundances in the ocean (Wu et al., 2000). The depletion of phosphate by
phytoplankton and microbial use in the upper photic zone can result in a lower stoichiometric P use
(relative to that of Co here), and could thus induce the positive trajectory of dCo:P through microbial loop
remineralization by effectively lowering the dCo:P denominator. There is biochemical evidence to
support this phenomenon, where many phytoplankton, including cyanobacteria and diatoms, decrease
their P stoichiometry by sparing (conserving) phosphate intracellularly through the substitution of
sulfolipids for phospholipids in their membranes. This effectively lowers their phosphate requirement and
deviates from Redfieldian N:P stoichiometries (Van Mooy, 2006). In addition to lipid measurements, the
biosynthesis proteins for sulfolipids also increase in abundance due to P scarcity in the North Pacific
Ocean (Saito et al., 2014). As a result, the marine cyanobacteria that dominate the oligotrophic gyres of
the Atlantic having a large dynamic range for their phosphate stoichiometry: cellular composition of
axenic cultures of *Prochlorococcus* and *Synechococcus* showed 3.8-5.6 fold decreases in C:P ratios
between P-replete and P-limited conditions (C:P ratios limited: 46, 50 and 63 and replete: 179, 290, and
301 for MED4, 8102, and 8103, respectively) (Bertilsson et al., 2003). In addition, there is field evidence
supporting this P-sparing phenomenon in the North Atlantic cyanobacterial populations, including a 2-
fold decrease in *Synechococcus* P content in the low-nutrient anticyclonic eddy in the Sargasso (Twining
et al., 2010), and with picocyanobacteria showing C:P ratios elevated by as much as 10-fold over Redfield
values in surface and deep (142 m) samples (Grob et al., 2013). In summary, this replacement of sulfur
for phosphorus within membranes would increase the upper photic zone dCo:P stoichiometry by
depressing the denominator.
The second and third processes (increased biotic cobalt demand and substitution of cobalt for
zinc) that can explain accelerating dCo:P stoichiometries are closely related, and likely occur
simultaneously. The preferential use of P explanation described above does not appear to entirely explain
the dCo:P relationships because particulate cobalt, likely reflecting the microbial community (filtered by
0.2 μm membranes), both increased in absolute concentration towards the surface (Fig. 7(b)) and had a
high pCo:pP biomass ratio of ~300 μM:M (Fig. 7(d)). Notably, these particulate values were similar to
the dissolved phase profile-regression results (Fig 7(c); Fig 4(c), (f)), and were roughly an order of
magnitude higher than the aggregate-regression reported above, implying the dissolved phase
stoichiometry was reflecting a high cellular content of Co in the protoplasm in the upper water column of
this region. Moreover, while the particulate reservoir of cobalt is generally a small fraction of total cobalt
reservoir (defined as dCo + pCo), in the upper 70 m pCo was typically greater than 10% of the cobalt
reservoir due to high microbial activity, increased Co demand, and drawdown of dCo (Fig. 7(f)).
These results were also consistent with additional data collected across the broad lateral gradients
of the North Atlantic zonal GA03 transect, where elevated pCo:pP in labile particulate matter was
observed in the low phosphate mid-subtropical gyre region, and phytoplankton cells were observed to
have a high stoichiometry of ~400 μmol mol$^{-1}$ Co:P, as measured by synchrotron X-ray fluorescence
(SXRF) at Station 2011-16 (Twining et al., 2015) (their Fig. 11c). As a result, the large relative size of the
pCo reservoir will rapidly impose changes on the dCo:dP ratio through the continual activity of the
microbial loop (uptake and grazing/lysis) that is known to turn over the entire euphotic zone small particle
reservoir (represented by picoplankton) every two days in oligotrophic regions (Cavender-Bares et al.,
1999; Mann and Chisholm, 2000; Vaulot, 1995). As the picoplankton and other microbial populations





that dominate the subtropical gyres are continually grazed and lysed, the particulate pCo reservoir is
released back to the dCo phase. As the percentage of pCo to dCo increases towards the surface, the
particulate phase gains additional 'leverage' with which to alter the stoichiometry of the dissolved phase.
**3.2.4 Connecting Metal Distributions to Metaproteomic Metalloenzyme Distributions**
6          Together these results present imply high Co demand in the upper photic zone in the surface
Atlantic Ocean. Why might this higher Co use occur? The enhanced dCo:P and pCo:P observed in the
upper photic zone likely reflects an increased cobalt requirement in the microbial community. While the
marine cyanobacteria *Prochlorococcus* and *Synechococcus* both have an absolute requirement for Co,
where they cannot survive without it nor can they substitute Zn for Co (Saito et al., 2002; Sunda and
Huntsman, 1995), this absolute requirement appears to be a relatively minor component of their cellular
cobalt quota. Additional biochemical functions for Co have been hypothesized, in particular the use of
cobalt in alkaline phosphatase and carbonic anhydrase (Jakuba et al., 2008; Saito et al., 2002; Saito et al.,
2003). The alkaline phosphatase enzyme appears to be particularly important in the Atlantic oligotrophic
gyres where soluble reactive phosphate is extremely low (Duhamel et al., 2010; Dyhrman et al., 2002;
Jakuba et al., 2008), and low phosphate availability causes an increase in the biosynthesis of this enzyme
in order to allow phytoplankton to liberate phosphorus from the dissolved organic phosphate (DOP)
chemical reservoir (Dyhrman et al., 2012). Indeed, there have even been reports that the activity of
organic phosphorus acquisition may be constrained by zinc availability: recent field studies in the North
Atlantic observed stimulation of alkaline phosphatase activity through the addition of zinc (Mahaffey et
al., 2014).
22          There are two isoforms of the alkaline phosphatase enzyme, the zinc PhoA and recently
characterized iron-calcium PhoX (Duhamel et al., 2010; Shaked et al., 2006; Yong et al., 2014),
(previously thought to be a calcium-only enzyme (Kathuria and Martiny, 2011)). While PhoA is known to
be a zinc metalloenzyme in model organisms (Kim and Wyckoff, 1991), cobalt has been demonstrated to
substitute for the catalytic center in the hyperthermophilic microbe *Thermotoga maritima*
(Wojciechowski et al., 2002). It is unknown at this time if this cobalt-zinc substitution can occur in
marine microbes: the metal center of marine cyanobacterial PhoA has yet to be determined under natural
conditions in the laboratory or field environment. While the PhoX isoform's use of iron has been
hypothesized to lessen its dependence on PhoA in iron-rich waters (Mahaffey et al., 2014), PhoA was
observed to be more prevalent in *Synechococcus* proteomes under low phosphate relative to PhoX even in
replete iron conditions, implying PhoA could be particularly important for DOP utilization (Cox and
Saito, 2013). Also consistent with the use of Zn (and perhaps Co) was a lower abundance of PhoA at low
Zn while still at low P availability, implying that the expression of PhoA was co-induced by low P and
high Zn (Cox and Saito, 2013). The influence of low Co and P has not yet been explored in marine
cyanobacteria. The dissolved iron abundances of these stations in the Central North Atlantic were
elevated due to aeolian deposition with near surface towed samples being greater than 0.6 nM between
stations 11-12 and 11-20 (Hatta et al., 2015), although excess strong iron organic ligands were detected
throughout these regions as well (Buck et al., 2015), potentially reducing iron bioavailability.
40          We hypothesize that this elevated Co use in the upper water column is being driven by PhoA
alkaline phosphatase. To support this hypothesis, we present novel metaproteomic data from samples
taken at the Bermuda Atlantic Time-series Study station in the North Atlantic Subtropical Gyre during
April of 2015 (the same location as GEOTRACES station 11-10) that showed high abundances of
alkaline phosphatases (PhoA) in the upper euphotic zone (Fig. 8). Two distinct cyanobacterial alkaline





phosphatases were detected, both the PhoA isoform, from *Prochlorococcus* and *Synechococcus* species,
corresponding to sequences from Atlantic isolates NATL1 and WH8102. The *Synechococcus* PhoA
isoform was more abundant in the upper photic zone, while the *Prochlorococcus* PhoA showed a
maximum at a depth of 82 m, consistent with their depth distributions of marine cyanobacteria in this
region (Olson et al., 1990). Since PhoA is a metalloenzyme with two zinc atoms per protein, these
metaproteomic results imply an increasing need for a divalent cation, zinc or perhaps cobalt if substitution
was occurring, to populate this enzyme in the upper photic zone. Note that these protein profiles are
relative abundance units of normalized spectral counts. Future analysis by calibrated targeted
metaproteomics will allow protein concentrationss and their metals content to be estimated (Saito et al.,
2015; Saito et al., 2014). The PhoX iron-calcium isoform was not detected in the water column in this
preliminary analysis, although this negative result should not be interpreted as the protein being absent
from the ecosystem since it could reflect lack of matching PhoX sequences or annotations in our database.
While this BATS metaproteome dataset was geographically and temporally different from that of the
NAZT section, it was within the North Atlantic subtropical with its characteristically low phosphate
abundances. The sulfolipid biosynthesis protein (UDP-sulfoquinovose) also showed a surface maximum
at this BATS station (Fig. 8), demonstrating that this phosphate sparing mechanism was being engaged
and that we would also expect decreases in cellular phosphate quotas in the marine cyanobacteria as
described above. The lower phosphate inventory of the North Atlantic subtropical gyre versus the South
Atlantic subtropical gyre could also explain the differences in dCo:P stoichiometry maxima observed
between basins (Fig 4(c), (f)), where increased P scarcity could result in enhanced dCo:P stoichiometries
through the three processes described above.

22        Could zinc have been scarce enough to encourage cobalt-zinc substitution within metalloenzymes

such as alkaline phosphatase? Zinc can be exceedingly scarce in the upper photic zone due to
phytoplankton uptake and export, particularly in the subtropical gyres (Bruland and Franks, 1983). To
examine this possibility, we compared the distributions of dissolved cobalt, zinc, and cadmium (Cd can
substitute for Zn in diatom carbonic anhydrases; (Lane et al., 2005)) in the center of the North Atlantic
subtropical gyre, again at USGT11 stations 14, 16, and 18 (Fig. 9(a)-(c)). While dissolved Zn was the
most abundant of the three at intermediate depths (18-fold more dZn than dCo at 1000 m, Fig. 9(a)), it
became depleted within the upper 100 m to the extent that its concentrations were reduced to less than
two times that of Co (ratios of dCo:dZn are greater than 0.5, Fig. 9(g)). Dissolved Cd was so depleted in
the photic zone that dCo actually became 50-fold more abundant than dCd in the photic zone (Fig. 9(h)),
and the dCd was typically more than 100-fold lower than dZn in the upper euphotic zone (Fig. 9(i)).
Dissolved zinc and cadmium are also typically bound by strong organic complexes in the oceanic
euphotic zone (Bruland, 1992, 1989; Ellwood, 2004), which would greatly reduce the abundance of their
inorganic species and their resultant bioavailability to phytoplankton as observed in culture studies
(Sunda and Huntsman, 1995; Sunda and Huntsman, 2000).

37        Development and application of new metalloproteomic techniques (Aguirre et al., 2013) could

determine if cobalt can substitute for zinc within the PhoA metalloenzyme of the abundant cyanobacteria
*Prochlorococcus* and *Synechococcus* that dominate the oligotrophic euphotic zone when zinc and cobalt
become similar in abundance, consistent with observations of a cobalt-PhoA in a hyperthermophillic
bacterium (Wojciechowski et al., 2002). This comparison to proteomic results also demonstrates the value
in "Biogeotraces" efforts that aim to connect nutrient and micronutrient distributions directly with the
proteins that require them, as well as with additional biochemical, molecular, and cellular information
about the resident biota.




**3.2.5 Excess Zn Uptake in the Lower Photic Zone Creating dZn Convex and dCo Concave Kinks**

Interestingly, dZn and dCd have concave kinks when plotted against phosphate in this region (Fig. 9(d, (e)). This is in contrast to the convex kinks observed above in dCo:P space. It has been previously suggested that differences in the relationships of Co, Cd, and Zn relative to phosphate in the Ross Sea and North Pacific are indicative of variations in phytoplankton metal usage (Saito et al., 2010; Sunda and Huntsman, 2000). One explanation for this phenomenon is that there is excess biological uptake (defined as uptake in excess of the cellular biochemical requirements) at the base of the euphotic zone, resulting in Zn and Cd becoming rapidly depleted towards the surface to concentrations approaching Co (Saito et al., 2010). This depletion of Zn and Cd can then create conditions amenable to Co substitution in the upper euphotic zone. This excess Zn uptake and Co substitution scenario seems particularly plausible in these oligotrophic Atlantic Ocean gyre locations as well, leaving Co to have an important nutritional role and high stoichiometric values in the upper water column of this region. Cellular Zn:P values in individual phytoplankton cells across the GA03 North Atlantic transect were also measured by SXRF. Zn:P ratios were generally elevated near continental margins, and the lowest values were observed in the mid-subtropical gyre at station 2011-16 where cellular Co:P was elevated, consistent with Co substitution for Zn use (Twining et al., 2015)(their Fig. 11F). It is noteworthy that these depletion and kink features are occurring much deeper in the Atlantic subtropical gyres than in the Ross Sea and North Pacific, due to the deep euphotic zones created by very low biomass and high light transmission, and with nutrients supplied primarily by slow vertical diffusion processes.

**3.2.6 Comparison of Field Ecological Stoichiometries to Cellular Quotas and Implications for Biological Use of Biochemical Substitution**

The range of dCo:P stoichiometric values estimated for the aggregate and profile regressions for the North and South Atlantic datasets were at the low- and mid-range of the measured cobalt cellular quotas in phytoplankton grown at very low zinc abundances, respectively. Sunda and Huntsman reported Co:C quotas for the coccolithophore *Emiliania huxleyi*, the diatoms *Thalassiosira pseudonana* and *Thalassiosira oceanica*, and the cyanobacterium *Synechococcus bacillaris* (Sunda and Huntsman, 1995). When converted to Co:P, using an assumed C:P Redfield ratio of 106:1, the quotas over increasing cobalt and scarce $Zn^{2+}$ ($10^{-13}$M) ranged from 77-2713, 42-1314, 284-2120, and 8.5-151 µmol Co mol$^{-1}$ P, in the order of phytoplankton listed above. When zinc concentrations increased in those experiments, Co quotas decreased by several orders of magnitude in the first three eukaryotic phytoplankton strains, with no Zn quota data available for *Synechococcus*. Unfortunately, none of these culture experiments were conducted under P-limiting conditions that would induce phosphate sparing mechanisms and result in an enhanced Co or Zn stoichiometry. In a separate study, the coccolithophore *Emiliana huxleyi* was found to have a 16% increase in Zn cellular content when switched from growing on inorganic phosphate to organic phosphate (Shaked et al., 2006). However it is difficult to compare these results to the cyanobacteria that tend to dominate the Atlantic Ocean subtropical gyres since many cyanobacteria appear to have little demand for zinc when grown in inorganic P conditions (Saito et al., 2002; Sunda and Huntsman, 1995), although *Synechococcus* does show enhanced growth with zinc in media which includes organic P (Cox and Saito, 2013). Based on this comparison and the discussion above, we interpret that there may well have been significant substitution of Co for a combined Zn/Co requirement, particularly in the upper water column where dZn was roughly equivalent in concentration to dCo, assuming the enzyme(s) are capable of such a substitution.





### 3.2.7. The Accelerating Co Stoichiometry Phenomenon in the Context of Redfield Theory

The accelerating dissolved Co stoichiometry is a surprising feature that likely reflects an increasing influence of a high pCo quota on the dissolved reservoir towards the sunlit surface waters. To make sense of this we can reflect on Redfield et al.'s early writing on the dissolved and particulate C, N, and P sharing stoichiometric ratios, where they wrote: "Elements are withdrawn from sea-water by the growth of marine plants in the proportions required to produce protoplasm and are returned to it as excretions and decomposition products of an equally specific nature. … Since the elements required for the construction of protoplasm are drawn from the water in proportions which have some uniformity, they are distributed in somewhat similar patterns by the biochemical circulation." (Redfield et al., 1963). In this writing Redfield et al. not only emphasize a general ("statistical") uniformity of stoichiometry, but also a bidirectional flow of nutrients between dissolved and particulate phases, and its subsequent influence on seawater composition. The often observed stoichiometric equivalence in dissolved and particulate phases thus requires an implicit ability of these phases to materially exchange with each other through continual cellular uptake, grazing and lysis recycling, and remineralization processes to such an extent that the dissolved and particulate stoichiometries converge on identical ratios. The small amount of material that escapes an oligotrophic euphotic zone as export flux can then act within an important gradual winnowing process where stoichiometric excesses are removed from the dissolved phase into the particulate phase and remineralized below where they may have a minor influence on the larger preformed mesopelagic inventories.

Cobalt, as one of the scarcest of inorganic nutrients, provides an interesting counter-example to Redfield's abundant macronutrients. In oceanographic contexts while there is increasing evidence that there can be some regional variability in Redfield's stoichiometric ratios, these variations are relatively small (e.g., less than two-fold (Martiny et al., 2013)) when compared to the large multiple order of magnitude potential plasticity observed in metal usage in culture experiments (Sunda and Huntsman, 1995b, 2000, 1997). Yet for trace metal micronutrients such as Zn and Cd, the linear relationships between those metals and macronutrients implies a consistency (or an averaging) of stoichiometries in the oceans. In comparison, the large variability in cobalt ecological stoichiometry discussed here appears to be unusual. The situation for cobalt is extreme: not only are the dissolved Co:P ratios so variable as to make a single uniform oceanic ratio difficult, but they span more than an order of magnitude, and as described above, accelerate towards the surface. Such plasticity is likely enabled by the biochemical substitution strategies deployed by euphotic zone phytoplankton described above for Co and Zn, and the stoichiometry of these elements has been unequivocally demonstrated in the laboratory to be able to shift considerably (Sunda and Huntsman, 1995). If the considerably lower aggregate regression stoichiometries described above reflect much lower biochemical requirements in the base of the euphotic zone, it seems likely that the Atlantic with its particularly low phosphate availability results in a diversity of cobalt stoichiometries from the base of the euphotic zone (where P is abundant) to the surface where P scarcity results in the three mechanisms described above (see Section 3.2.3) simultaneously contributing to elevated Co:P. As a result, we are able to observe the pull of the upper photic zone on the biological stoichiometry of the dissolved phase stoichiometry and its distinct acceleration towards the surface. The acceleration of Co:P towards the surface was also supported by SXRF quota data on three stations on the GA03 North Atlantic transect, where Co quotas in cells were 2-4 fold higher in the upper mixed layer compared to the deep chlorophyll maximum, and reflected the largest depth quota difference of all trace metals studied in this region (Twining et al., 2015) (their Fig. 9 and Table 4).



An important general stoichiometric lesson that we can learn from cobalt is that the coupling
between the dissolved and particulate phase stoichiometries is not instantaneous, with each phase having
an inertia related to the size of its inventory and the extent of exchange between phases. The small size of
cobalt's water column inventory, and the potential for its stoichiometry to change greatly in response to
more abundant nutrients such as P and Zn, erodes away at its limited inertia and results in its acceleration
to catch up with the particulate phase.
7        The reader might have noticed that one piece of data is missing in this story: we would expect the
lower euphotic zone particulate phase to also show lower Co:P stoichiometries associated with
phytoplankton (growing in abundant Zn and P) and resultant lowered cellular particulate quotas, as is
clearly observed in the dissolved phase (Table 1) and in culture studies (Sunda and Huntsman, 1995).
However, these deeper phytoplankton stoichiometries appear to be masked by substantially higher Co:P
stoichiometries associated with microbial manganese oxide particles that do not appear to communicate
back with the dissolved phase (Fig. 7(e)), effectively acting as a one-way trip into the particulate phase.
This provides an opportune segue to the mesopelagic and deep ocean and the unusual negative dCo:P
stoichiometries observed therein.
**3.3 Evidence for Mesopelagic Scavenging of Cobalt in the Atlantic Ocean**
18       The cause of cobalt's small marine inventory is often attributed to be the result of scavenging
processes that continually remove dissolved cobalt from the water column. The evidence for this process
is limited to several field and laboratory radiotracer experiments that point to the co-oxidation of cobalt
within manganese oxide particles below the photic zone (Lee and Tebo, 1994; Moffett and Ho, 1996;
Tebo et al., 1984), as well as interpretation of vertical profiles with reduced cobalt at intermediate and
deep depths (Noble et al., 2008; Noble et al., 2013; Saito et al., 2010; Saito and Moffett, 2002). This
production of Mn oxide phases is a biological process where manganese oxides precipitate directly onto
the cell surface of manganese oxidizing bacteria (Cowen and Bruland, 1985), and hence could largely
decouple Mn and the co-precipitated Co from phosphate as these metals are largely incorporated into the
mineral phase rather than microbial biomass (see schematic in Fig. 4). Yet it was also recently suggested
that scavenging may not be an important process for dissolved cobalt in the oceans, and that instead
differences in deepwater concentrations are controlled by physical circulation and remineralization
processes (Dulaquais et al., 2014b). In general the notion of "hybrid-type" elements that possess both
nutrient-like and scavenged behaviors, including the metals Fe, Cu and Co, is relatively new (Bruland and
Lohan, 2003; Noble et al., 2008; Saito et al., 2010), and this large dataset provides a useful opportunity to
provide evidence and discussion of this phenomenon.
34       The profile-based regressions employed above for upper water column processes as well as
particulate metal datasets can provide insight into the scavenging process. As noted above, the profile
analysis using 5-point moving two-way linear regressions identified numerous depth intervals with
negative linear relationships between Co and phosphate in both the North and South Atlantic Ocean (Fig.
3.; red symbols and lines), that are distinct from the positive slope relationships attributed to uptake and
remineralization processes described above (also exceeding a selected threshold correlation coefficient (r)
of $\geq |0.7|$). These negative slopes can be located with their dCo concentration profiles (Fig. 5, red
symbols) and their magnitude examined with depth (Figs. 6; red symbols). Note that the correlation
coefficients of negative slopes are also negative (e.g., < -0.7), and only data with r-values above the
threshold are presented.
44       These negative relationships are intriguing in that they deviate from the idealized downward





(vertical) vector for scavenging (Fig. 4(a) inset), with measured slopes that generate "southeast" vectors
in both the North and South Atlantic (Fig. 4(a), (c), red vectors). These negative vectors imply the
removal of dCo simultaneously with an addition of phosphate from the water column. It is difficult to
envision a single process that can create this effect; however, the addition of two vectors makes this
feasible: a positive remineralization vector plus a near vertical scavenging vector can reproduce both the
negative vectors and their decreasing slope (becoming increasingly vertical) with depth as the
remineralization contribution dissipates and approaches zero. This is demonstrated in a revised schematic
(Fig. 10(c)) and vector addition diagrams (Fig. 10(a)-(b)) that use measured values from this study.
Vector "end-members" for remineralization of euphotic zone biomass and Mn oxidation were calculated
using measured pCo and pP from Go-Flo bottle samples for the upper water column and McLane pump
samples for deeper values at station USGT11-18 (Fig. 10(a)). These mesopelagic and deep Co
stoichiometric values were relatively consistent across the North Atlantic basin as shown in aggregate
particle concentrations (Fig. 11(a)-(b); McLane pump samples) and as ratios (Fig. 11(c)-(d)), with
pCo:pMn ratio of $0.013 \pm 0.002$ M:M and a pCo:pP ratio of $1840 + 640$ μM:M (>400 m; excluding the
North American shelf and nepheloid layers, n = 129). Notably, these deep pCo:pP stoichiometries were
considerably higher than the dissolved and particulate stoichiometries associated with the euphotic zone
likely due to the accumulation of Co within Mn oxide phases, with cobalt being ~1% the molar
abundance of manganese in these deep particles consistent with it being a minor component of the
manganese oxide phase. Lithogenic corrections were not included here for pCo and pP since these
elements were weakly explained by lithogenic contributions in the North Atlantic (Ohnemus and Lam,
2015). While the vector addition exercise is a comparison of two different filter pore sizes that were used
in order to capture ratios for each depth region as described in the methods (0.2 vs 0.8 μm in bottle versus
pump particles), it is clear from Fig. 11(c) that even within the pump particulate dataset the pCo:pP
decreases dramatically towards the surface. Also this deep pCo:pMn ratio was much lower than the 0.1-
0.4 (M:M) ratio observed in the photic zone due to opposing trends of increased pCo due to biological use
and decreased pMn due to photoreduction of Mn oxides and limited biological use.
Using these representative particulate values, addition of example Mn oxide and remineralization
vectors was able to reproduce the southeast negative slope vectors found throughout the profile-based
regression analysis in the mesopelagic ocean (Fig. 10(b) versus Fig. 3(b), (f) and Fig. 4(a), (d). Note that
the vectors magnitudes were chosen for demonstration purposes here (2-fold for Mn oxidation, 1/5th for
remineralization to allow for attenuation of the remineralization flux at depth), but in the water column
one can envision a gradual transition between these two vectors (as visualized in Schematic Fig 10(d)):
from remineralization dominating at the surface and Mn oxidation dominating at depth (note that uptake
is not included since it withdraws from the dissolved phase while remineralization adds to it). This trend
can also be summarized by Eqn. 1, where the balance shifts from being dominated by the combined
uptake and remineralization terms *(- ρ + remin)* in the upper water column to being dominated by
scavenging removal term *(Scav)* transitions with increasing depth as sinking organic matter is depleted by
remineralization. This scenario is consistent with the range in observed slopes, where the negative
scavenging regressions tended to increase in steepness with depth (Figs. 3D, H), as well as in the vector
diagrams (Fig. 4(a), (d)) and histogram distributions of frequency of dCo:P slopes (Fig. 4(b), (e)).
$$\frac{dCo}{dt} = (-\rho + remin) - Scav + diffusion \qquad (1)$$
Hence we interpret these negative dCo:P relationships as evidence of an increased influence of





the scavenging removal process below the euphotic zone within each water mass. The placement of these
negative slope regions in Co:P space is also consistent with the "cobalt curl" away from the Co:P positive
linear relationship that is common in the upper water column (see Section 3.1 above) that we have
previous discussed (Noble et al., 2008; Saito et al., 2010). Implicit in this discussion is that the Mn oxide
solid (microbial) phases are not subject to a significant extent of grazing or viral lysis that would release
their constituents back to the dissolved phase, effectively creating a "one-way" trip for dCo into the
particulate phase. While little is known about mesopelagic grazing processes, these data appear to be
consistent with a net transport into the particulate phase, with no evidence for a north or northwest Mn
oxide remineralization vector. These scavenging signals co-occurred with distinct water masses identified
by OMPA analysis (optimum multiparameter analysis), implying that these scavenging processes are
being integrated on decadal-to-century timescales of deepwater circulation processes within the ocean
interior (Noble et al., submitted). Specifically, negative slope water masses were found to be in the
Denmark Straits Overflow Water/Antarctic Bottom Water/Iceland Scotland Overflow Water
(DSOW/AABW/ISOW) and Classical Labrador Seawater (CLSW; Fig. 2) water masses both of which
have long deepwater transit times (Jenkins et al., 2015).
16          One interesting aspect of these analyses was the high degree of depth and spatial variability of the
dCo:P relationships. In particular, there were regions of positive and negative Co:P relationships
*vertically interspersed* within numerous individual vertical profiles (Figs. 5-6; e.g., stations USGT10 9-
11). The presence of water masses with positive dCo:P slopes well below the photic zone was also
intriguing. This variability is typically found below the "cobalt-cline" and continues into the deep ocean,
and can be attributed to either temporal variability in export and remineralization and/or the horizontal
advection and interweaving of remineralization signals within water masses in the vertical profile.
Alternatively, if scavenging processes were to be reduced in a parcel of water for some reason, a
remineralization signal could emerge, reflecting a shift in the balance of (vector) contributions. There is
limited understanding regarding the controls on Mn oxidation, and hence it is difficult to imagine a
mechanism for repression of scavenging at this time, although presumably Mn oxidation microbial
activity is coupled to organic matter flux and hence its overall contribution would also dissipate with
depth. It may be possible to use this profile based regression analysis of dCo and P to generate an ability
to detect spatial and temporal variability in deep export and remineralization events by their deep positive
Co:P relationships, if background scavenging rates could be constrained.
31          Analysis of specific vertical profiles of the particulate Co and Mn data provides further evidence
for Co scavenging in the mesopelagic, and implicates manganese oxides as the responsible phase and for
the transitioning of major processes with depth (Fig. 12). Comparing the dissolved and particulate phases
of example vertical profiles in the Eastern North Atlantic at Stations 7 and 10 (USGS-2010), distinct
zones of correlations were observed between dCo, $PO_4^{3-}$, pCo and pMn, above and below the 400 m depth
horizon (Fig. 12(a), (g)). The upper ocean showed a linear correlation of dissolved cobalt with phosphate
(Fig. 12(b) and (h), black symbols), indicative of phytoplankton uptake and remineralization of sinking
material as described above. Below the 400 m, a correlation between particulate Co and particulate Mn
emerged, consistent with the scavenging process influencing both elements through incorporation into the
biomineral manganese oxide (Fig. 12C and I, red symbols). For station GT10-07, pMn and pCo showed
this linear relationship throughout the deep ocean (from 400 m to 4500 m), but with absolute values of
both particulate metals decreasing with depth implying more active Mn oxide formation in the upper
mesopelagic (Fig. 12(c), (f)). An inverted L-shaped relationship was observed in some cases between
both pMn and pP, and pCo and pP, (Fig 12(d), (e), (k)) due to deeper particles having higher metals





associated with the Mn oxide phase and lower phosphate than euphotic biogenic particles. These
observations can be generalized across the North Atlantic GA03/3_e section by comparison of particulate
Co, Mn and P and applying this 400 m horizon. Correlations of pCo with both pMn and pP were observed
below 400 m (Fig. 13; $r^2$ 0.81 and 0.47), with enriched pCo:pP relative to the shallower <400 m zone
consistent with its accumulation in microbial manganese oxidizing bacteria.
6       Together, these dissolved and particulate datasets demonstrate the overall nature of the major
competing processes on cobalt distributions, where the processes of phytoplankton uptake and
remineralization dominate in the euphotic zone and just below it, versus the scavenging process that takes
precedence as the remineralization signal subsides, as shown in schematic Fig 10D. The biological pump
processes dominate in the upper water column but should attenuate rapidly in the mesopelagic
comparable to the power law decay of carbon remineralization as described by Martin et al. (Martin et al.,
1987), and the export flux contribution is likely to vary geographically due to the episodic nature of
phytoplankton blooms. Simultaneously, the contribution of scavenging likely increases below the upper
euphotic zone as light subsides and photoreduction of manganese oxides ceases (Sunda and Huntsman,
1994), yet little is known about what might regulate the extent of bacterially catalyzed manganese
oxidation, but could include bacterial activity, organic matter, manganese, pH, and/or dissolved oxygen
availability (Johnson et al., 1996; Morgan, 2005; Tebo et al., 1984). Indeed, Moffett and Ho observed a
large dynamic range of 300-fold difference in manganese oxidation rates, as a percentage of tracer
oxidized per hour, between a coastal estuary and the oligotrophic ocean (Moffett and Ho, 1996). The
relative contribution of these two processes should invert near the maximum of the dissolved cobalt
profile, contributing to its characteristic sharp peaks at these intermediate depths (Fig. 3). With this
competition between vertical processes, the depth of the cobalt maximum could vary with the extent of
vertical export and Mn oxide production. The balancing act between these two major processes is
apparent in the variability of individual profiles, demonstrating the complex hybrid nature of the
dissolved cobalt profile. Scavenging of dCo appears to be a critical process in controlling the inventory of
cobalt in the oceans, and this topic, including the estimate of overall scavenging influences during
horizontal advection through ocean basins, is further explored in Hawco et al. (*in prep*).
**3.4 Ocean Sections of Co* and dCo:P slopes.**
30       Two derived values of dissolved cobalt can be calculated across the entire North Atlantic Ocean
section and presented visually using Ocean Data View to provide large scale assessment of cobalt's
properties. The first of these is the dCo:P slope value calculated by the profile-regression approach
described above (representing the mid-point of linear regressions for successive groupings of five depth
points in each vertical profile). The second of these is a "nutrient-star" calculation similar to that used
originally for nitrogen and applied to other nutrients (Gruber and Sarmiento, 1997; Deutsch et al., 2001).
Here Co* represents a deviation from "Redfield" stoichiometric use, and was calculated using equation 2:
38                    $Co^* = dCo - QP$           (2)
where Q represents the assumed Co:P quota value of 237 μmol:mol was used based on aggregate pCo:pP
ratio in the upper 400 m (pump dataset), and an intercept of zero was assumed implying a basal
requirement in life for both Co and P. Both of these assumptions are debatable given cobalt's unusual
biochemistry as described above: The basal Q value is likely subject to the environmental conditions of
each region, especially the phosphorus content as described above, but appears reasonable compared to



basin-wide least square average of 150 for picoplankton (Twining et al., 2015). Selecting a "Redfield"
cobalt Q value is not trivial, since the stoichiometry of Co:P as described above can be highly variable,
hence this current effort should be considered preliminary.
4        The resulting Co* section was surprising in its smoothness with gradual transitions from the
surface to deep ocean, for an element with such small and dynamic inventory. Notable features include
low Co* values in the mesopelagic and deep ocean due to scavenging, with the OMZ region being lowest
despite being the location of a major Co plume. The Co* deficit observed within the subsurface could be
a useful indicator for the integrated influence of cobalt scavenging. The selection of this moderate Q
value results in Co* values that were considerably in excess of unity in the upper water column.
Obviously any shift in Q would shoal or deepen these features, and hence the accelerating stoichiometries
observed in the upper photic zone are problematic in deploying in a single derived Co* field.
12        A sectional visualization of dCo:P slopes was strikingly distinct from that of Co* with a
patchiness associated with distinct regions and depths. These patches were the regions of accelerating
slope identified in the profile based analysis described earlier. There were several salient trends apparent
in this section. First, strong Co scavenging at the hydrothermal vents was readily apparent, caused by high
near-field cobalt concentrations being subjected to rapid losses without comparable (stoichiometric)
losses in phosphate. Second, the enhanced remineralization (positive) slopes, described briefly above,
were apparent below the OMZ of the Mauritanian Upwelling. This intriguing observation implies that
material sinking through the OMZ was prevented from degrading rapidly due to low oxygen, but below
the OMZ remineralization resumed. Alternatively, it could imply the remineralization of biomass from
within the OMZ that has higher pCo:pP quotas, as observed recently in the U.S. GEOTRACES Eastern
Tropical zonal transect (Ohnemus et al., in press) resulting in an acceleration of Co:P in the dissolved
phase. The archaea known to inhabit the ocean interior have high abundances of $B_{12}$ biosynthetic proteins,
supporting this notion of a deep biological Co demand and potential export (Santoro et al., 2015). Finally,
the elevated slopes observed at station 11-06 on the North American Atlantic shelf could indicate either
subducted surface waters with a highly elevated Co:P stoichiometry or evidence of remineralization of a
prior export event.
**4. Conclusions and Implications**
30        In this study the relationships of dissolved and particulate cobalt relative to phosphorus on zonal
sections of the North and South Atlantic were investigated and their implications for the ecological
stoichiometry and biogeochemistry of cobalt were described. In particular, the finer-scale structure of
dCo:P relationships was characterized by use of linear regressions on small subsets of data within each
vertical profile on the sections. The most prominent observations were that the dissolved cobalt
stoichiometry varied by more than an order of magnitude and that the sign of the relationships switched
from positive to negative in the mesopelagic. In the upper photic zone, an acceleration of these
stoichiometries was observed in the dissolved phase due to a combined influence of phosphate scarcity
and its biochemical influence on cellular P use, as well as increases in Co use upon Zn depletion and
within the cyanobacterial alkaline phosphatase metalloenzyme, as supported by metaproteomic data. In
the mesopelagic, the observance of negative dCo:P relationships coincided with adherence of the
particulate cobalt phase with the particulate manganese phase providing direct evidence of the influence
of manganese scavenging upon dissolved cobalt. The biogeochemical cycling of cobalt is interesting
when compared to Alfred Redfield's early consideration of the connection between dissolved and
particulate phases through oceanic "biochemical circulation". With the smallest inventory of any required





nutritional element in the oceans and its potential for biochemical substitution, dissolved Co stoichiometries found in the oceans appear to be among the most dynamic of any element used by life. As a result, the coherence in stoichiometry between dissolved and particulate phases appears less of a duet as for other elements (N, P, Cd, Zn) than a tug-of-war for control of processes. As human use of cobalt grows exponentially with widespread adoption of lithium ion batteries, there is potential to disrupt cobalt's dynamic biogeochemical cycling and ecology in the oceanic euphotic zone.

**Acknowledgements**

This work was funded by the National Science Foundation as part of the U.S. GEOTRACES North Atlantic Zonal Transect program under grants OCE-0928414 and OCE-1435056 (to M.A.S), OCE-0928289 (to B.S.T), OCE-0963026 (to P.J.L) and support from the Gordon and Betty Moore Foundation (3782 to M.A.S.). We are indebted to the Captain and Crew of the R/V Knorr for their exemplary support on both GA03 and GAc01 expeditions, as well as chief scientists Bill Jenkins, Ed Boyle, and Greg Cutter, and the dissolved and particulate sampling teams. We also appreciate the support of the Captain and Crew of the R/V Atlantic Explorer, as well as support from the BATS group for assistance with McLane protein profile sampling.





1    **Table 1.** Ecological stoichiometries for dissolved cobalt, labile cobalt, and phosphate in the Atlantic Ocean and prior

2    studies. LCo refers to labile cobalt, all other values are total dissolved cobalt (dCo).

| Geographic Location | Study | Depth (m) | Co (pM) | ΔCo:ΔP ($\mu$mol mol$^{-1}$) | $r^2$ | n |
|---|---|---|---|---|---|---|
| North.Atlantic aggregate LCo | this study | 48 - 300 | n.d. - 48 | 23 | 0.82 | 156 |
| South Atlantic aggregate LCo | this study/Noble et al., 2012 | 48 - 300 | n.d. - 39 | 7 | 0.25 | 71 |
| North.Atlantic aggregate TCo | this study | 48 - 300 | 9 - 150 | 64 | 0.89 | 156 |
| South Atlantic aggregate TCo | this study/Noble et al., 2012 | 48 - 300 | 11 - 161 | 53 | 0.83 | 76 |
| Eastern North Atlantic | this study | 90 - 900 | 34 - 94 | 41 | 0.92 | 41 |
| Mauritanian Upwelling | this study | 48 - 425 | 26 - 157 | 48 | 0.83 | 53 |
| North Atlantic Subtropical Gyre | this study | 135-400 | 31-144 | 67 | 0.93 | 68 |
| South Atlantic Subtropical Gyre | this study/Noble et al., 2012 | 70 - 200 | 13 - 58 | 31 | 0.71 | 28 |
| Angola Gyre | this study/Noble et al., 2012 | 0 - 400 | 12 - 165 | 48 | 0.79 | 59 |
| North Atlantic Station 14 (2011) | this study; profile analysis | 40-136 | 17-45 | 320 | 0.71 | 5 |
| North Atlantic Station 16 (2011) | this study; profile analysis | 40-136 | 16-40 | 544 | 0.79 | 5 |
| North Atlantic Station 18 (2011) | this study; profile analysis | 40-137 | 13-38 | 491 | 0.51 | 5 |
| North Atlantic Station 20 (2011) | this study; profile analysis | 40-137 | 13-44 | 197 | 0.67 | 5 |
| N.E. Pacific (T5) | Martin et al., 1989 | 50-150 | 8 - 32 | 40 | 0.98 | 3 |
| N.E. Pacific (T6) | Martin et al., 1989 | 50-150 | 28 - 40 | 36 | 0.99 | 3 |
| N.E. Pacific (T8) | Martin et al., 1989 | 8 - 50 | 25 - 55 | 38 | 0.97 | 3 |
| Equatorial Atlantic | Saito and Moffett, 2002 | 5 | 5 - 87 | 560 | 0.63 | 14 |
| Peru Upwelling | Saito et al., 2004 | 8 | 21 - 315 | 275 | 0.96 | 11 |
| Ross Sea, Antarctica | Saito et al., 2010 | 5-500 | 19 - 71 | 38 | 0.87 | 164 |
| Subtropical Pacific (Hawaii) | Noble and Saito et al., 2008 | 0-300 | 3 - 52 | 29 | 0.63 | 33 |
| Subtropical Pacific (Hawaii) | Noble and Saito et al., 2008 | 0-250 | 11 - 47 | 37 | 0.91 | 19 |
| Southern Ocean (S1) | Bown et al., 2011 | 20-100 | 24 - 44 | 49 | 0.91 | 5 |
| Southern Ocean (S2) | Bown et al., 2011 | 15-120 | 7 - 26 | 44 | 0.99 | 5 |
| Southern Ocean (L4) | Bown et al., 2011 | 30-150 | 27 - 48 | 48 | 0.87 | 4 |

4





**Figure Captions**
Figure 1. Expedition tracks of the US North Atlantic GEOTRACES zonal transect (GA03/3_e; USGT10;
cruise number KN199-4, stations 1-12; and USGT11, KN199-5b stations 1-24) and the GEOTRACES-
compliant CoFeMUG South Atlantic Expedition (GAc01; KN192-5, 2007). Stations were numbered
sequentially from the beginning of each expedition (Portugal for GA03_e, Woods Hole for GA03, USA,
and Natal, Brazil for GAc01; respectively) with station numbers shown for selected stations. The North
Atlantic stations are described in later figures by the year and station number (e.g., 1101 for the 2011
expedition, station 01).
Figure 2. Total dissolved cobalt versus phosphate distributions observed across different regions in the
North (GA03/3_e) and South Atlantic (GAc01; left and right panels, respectively). Water masses were
identified by OMPA analysis (DSOW – Denmark Straits Overflow Water, ISOW – Iceland Scotland
Overflow Water, CLSW – Classical Labrador Sea Water, MOW – Mediterranean Overflow Water,
ULSW – Upper Labrador Sea Water, UCDW – Upper Circumpolar Deep Water, AAIW Antarctic
Intermediate Water, ISW Irminger Sea Water; Jenkins et al., 2014, their Table 1). In the South Atlantic
water masses correspond broadly to water masses as described in Saito et al., 2012, where UNADW
<2000m (Upper North Atlantic Deep Water), AABW >4000 in the western basin, and TDD (Two-Degree
Discontinuity) was the major contributor to both 2000-4000 m and >4000 m in the eastern basin of the
South Atlantic. Blue arrows indicate areas with steep dCo:P relationships.
Figure 3. Five point moving window linear regression analyses of dissolved cobalt versus phosphate
space. (a)-(d) All data in the North Atlantic GA03/3_e section. (e)-(h) All data in the South Atlantic
GAc01 zonal section. (b),(e) Data with a positive slope and r values greater than 0.7, after applying linear
regressions on groups of five vertically adjacent data points within each profile across the entire transect.
Solid blue lines represent the linear regressions superimposed on the data points analyzed. (c), (g) Data
with a negative slope and r values less than -0.7, after applying linear regressions on groups of five
vertically adjacent data points within each profile across the entire transect. Solid red lines represent the
linear regressions superimposed on the data points analyzed. (d), (h) Linear regression slopes versus depth
for the 5-point groupings, blue and red symbols represent positive and negative slopes, respectively.
Figure 4 (a) Schematic of idealized vectors for processes that influence dissolved cobalt. Vectors
measured by five-point moving window linear regression for (b) the North Atlantic zonal section, and (c)
the South Atlantic zonal section. Black and red colors correspond to the positive and negative slopes as
described in Fig. 3.
Figure 5. Vertical profiles of dissolved cobalt (pM) versus depth (m) from the North Atlantic GA03/3_e
section (black symbols) with stations listed above by expedition year (first two digits for USGT10 and
11) and station number (last two digits). Blue and red overlaid circles represent positive and negative
slopes by linear regression ($-0.7 < r > 0.7$), to indicate phytoplankton/remineralization and scavenging
processes, respectively.
Figure 6. Vertical profiles of dCo:P stoichiometry, as calculated by 5-point linear regressions (see
methods) for each of the GA03/3_e stations with stations listed above by expedition year (first two digits





for USGT10 and 11) and station number (last two digits). Blue symbols indicate positive slopes with
associated r values > 0.7. Red symbols indicate slopes with associated r values < -0.7. There was
significant geographical heterogeneity in stoichiometry. Most stations showed nutrient-like positive
slopes in the upper water column, and scavenged negative slopes in the mesopelagic and deeper depths.
Other variations with some stations showing only scavenged negative (red) slopes (1102, 1101),
particularly in the North American coastal region, while other regions showing alternating positive and
negative slopes likely indicative of subducted water masses (1106, 1124). Also, in the photic zone of
stations 1118, 1116, and 1114 there is an increase (an acceleration) of dCo:P stoichiometries towards the
surface. Some data points do not appear (e.g., stations 1118 and 1120) due to being off-scale and below
threshold.
Figure 7. Dissolved (dCo) and particulate (pP) cobalt and phosphate concentrations and ratios at
subtropical gyre Stations 14, 16, 18, and 20 from USGT-2011. (a) Phosphate profiles in the upper 350 m
for these four stations were low compared to all stations on the GA03/3_e expedition (small dots). (b)
Particulate cobalt (pCo) profiles (from the Go-Flo filters). (b) Particulate phosphate (pP) profiles (from
the Go-Flo filters). (d) dCo:P slopes generated by profile-regression analyses towards the surface, where
each depth represents the mid-point of 5 depths used in the profile-regression. (e) Ratios of pCo:pP
decrease towards the surface as they transition from being dominated by manganese oxide particles at
depth, but remain high relative to dissolved stoichiometries (Table 1) and culture studies at values of
~350-400 μmol/mol. (f) Percent of pCo of the cobalt inventory (dCo + pCo) revealed pCo to reach values
as high as ~10% in the upper euphotic zone, providing greater leverage for altering the dCo:dP slopes.
Figure 8. Profiles of (a) phosphate, (b) sulfolipid biosynthesis protein (UDP-sulfoquinovose), and (c)
alkaline phosphatase enzymes (PhoA isoform) at the Bermuda Atlantic Biological Time-series Study
station in April 2015. Proteins are in units of normalized spectral counts (sc). PhoA is generally
considered a zinc enzyme in model terrestrial organisms but has been found to be a Co enzyme in a
hydrothermal bacterium.
Figure 9. Zn, Cd, and Co and labile cobalt distributions (a)-(c), relationships with phosphate (d)-(f), and
as metal-metal ratios (g)-(i) in the central North Atlantic subtropical gyre at USGT-2011 stations 14, 16,
and 18. Dissolved cobalt is in blue and labile cobalt is red in panels (c) and (f). dZn and dCd were
depleted at the base of the euphotic zone resulting in dCo being ~40% of the abundance of dZn and 1-2
orders of magnitude more abundant than dCd within the euphotic zone. dZn and dCd have concave
relationships with phosphate, while dCo and LCo have convex relationships, implying faster biological
drawdown of use of dZn and dCd relative to phosphorus, and vice versa for dCo.
Figure 10. Vector addition demonstrating how negative dCo:P slopes can be generated by addition of
scavenging by Mn oxidation and remineralization of phytoplankton material. (a) Vector diagram
representing the uptake of dissolved cobalt and phosphorus into particulate Co and P for photosynthetic
and manganese co-oxidation processes. These vectors were generated using the measured pCo and pP
from Go-Flo bottle samples (67 m and 136 m) for the upper water column and McLane pump samples for
deeper values (420 m and 3000 m) at station USGT11-18. Solid vectors are represented as negative
vectors to portray the uptake into particles at each depth, a dashed vector portrays remineralization
releasing Co and P back to the dissolved phase. (b) Example addition of Mn oxidation vector and





phytoplankton remineralization that results in a negative vector as observed throughout the intermediate
depths in Figs. 3-6. (c). Idealized version of vector schematic, including the net mesopelagic vector. (d).
Idealized relative influence of processes on the dissolved distributions of cobalt and phosphate, using the
same color scheme as (c) and the euphotic zone in blue and mesopelagic in grey. The net vectors
summing the influence of all processes is on the far right, and is consistent with the shift from positive to
negative dCo:P slopes with depth calculated in Fig. 3.
Figure 11. Full depth distributions of (a) particulate cobalt (pCo), (b) phosphorus (pP), (c) ratios of
pCo:pP (inset 0-500 m depth), and (d) ratios of pCo and particulate manganese (pCo:pMn) from the
North Atlantic zonal GA03 section (McLane pump 0.8-51 µm filter samples).
Figure 12. Dissolved and particulate Co and associations with P and Mn from selected stations (USGT10-
7 and 10-10) from the GA03/3_e expedition. Dissolved Co is shown in (a), (g), particulate Co and Mn
profiles from McLane pump collected particle samples are shown in (f), (l), and comparison of dissolved
Co and dissolved phosphate, particulate Co and phosphate and particulate Co and Mn are shown (b)-(e);
(h)-(k). dCo and phosphate showed linear relationships above 400 m (b), (h), while pCo and pMn were
related below 400 m (c),(i), consistent with a transition between uptake and remineralization dominance
(<400 m) and scavenging by manganese oxides (>400m), and the profile vertical structure (a), (g).
Figure 13. Comparison of pump particulate Co, Mn and P in the North Atlantic (GA03/3_e) above (black
symbols) and below (red symbols) 400 m depth as evidence for scavenging of cobalt. (a) Higher pCo:pP
relationships are observed (160 µmol:mol Co:P) below 400 m likely due to the prevalence of Co
incorporation into Mn oxides as demonstrated by the high pMn:pP (b) and linear relationship between
pCo and pMn (c) observed below 400 m.
Figure 14. Comparison of derived variables to dissolved cobalt and phosphorus inventories in the zonal
portion of the U.S. North Atlantic transect (GA03). Ocean sections of (a) dissolved cobalt (pM), (b)
phosphate (µM), (c) dCo:P slopes (r $\geq$ |0.7|), and (d) Co* (with a Co:P stoichiometry of 237 µmol mol[-1]
based on the aggregate pCo:pP ratio in the upper 400 m).



1    Figure 1.

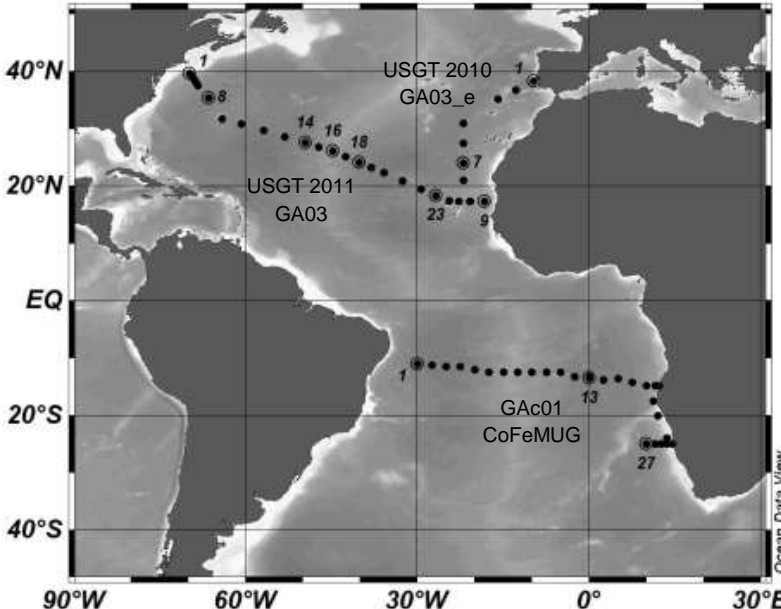



1    Figure 2.

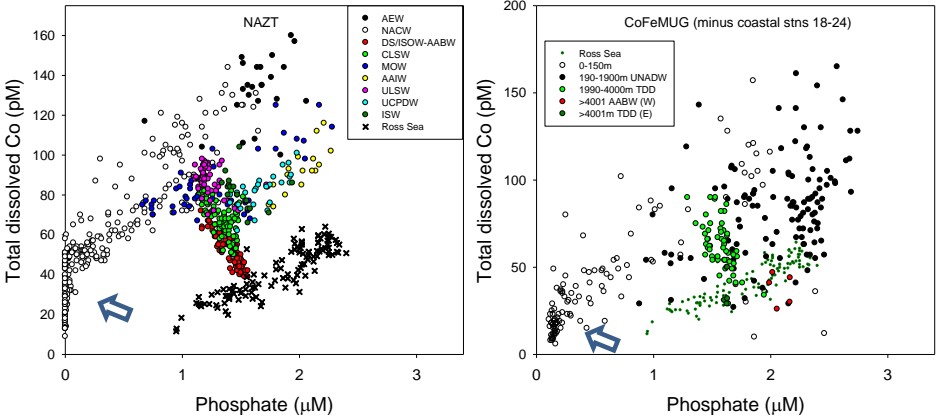





1    Figure 3.

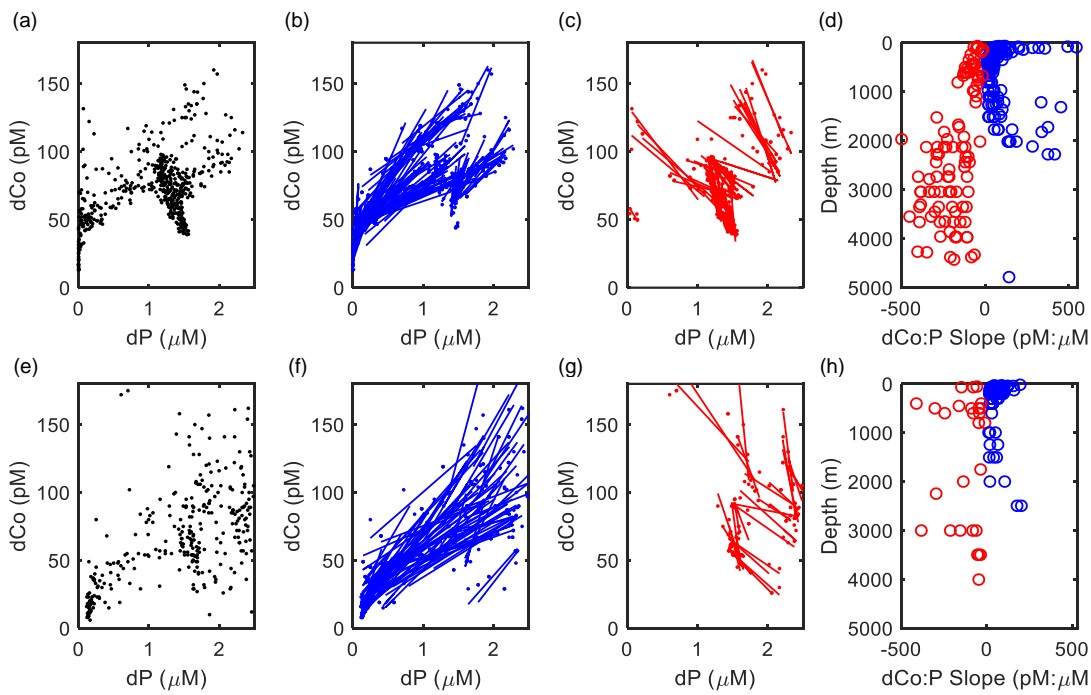





2    Figure 4.

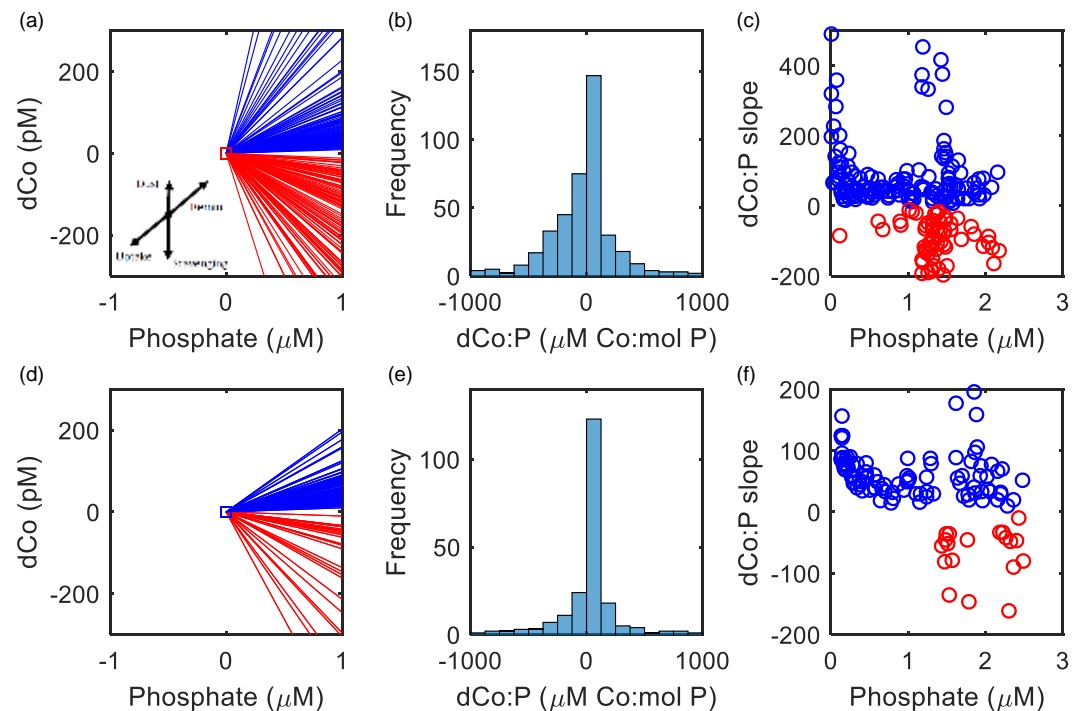





1    Figure 5.

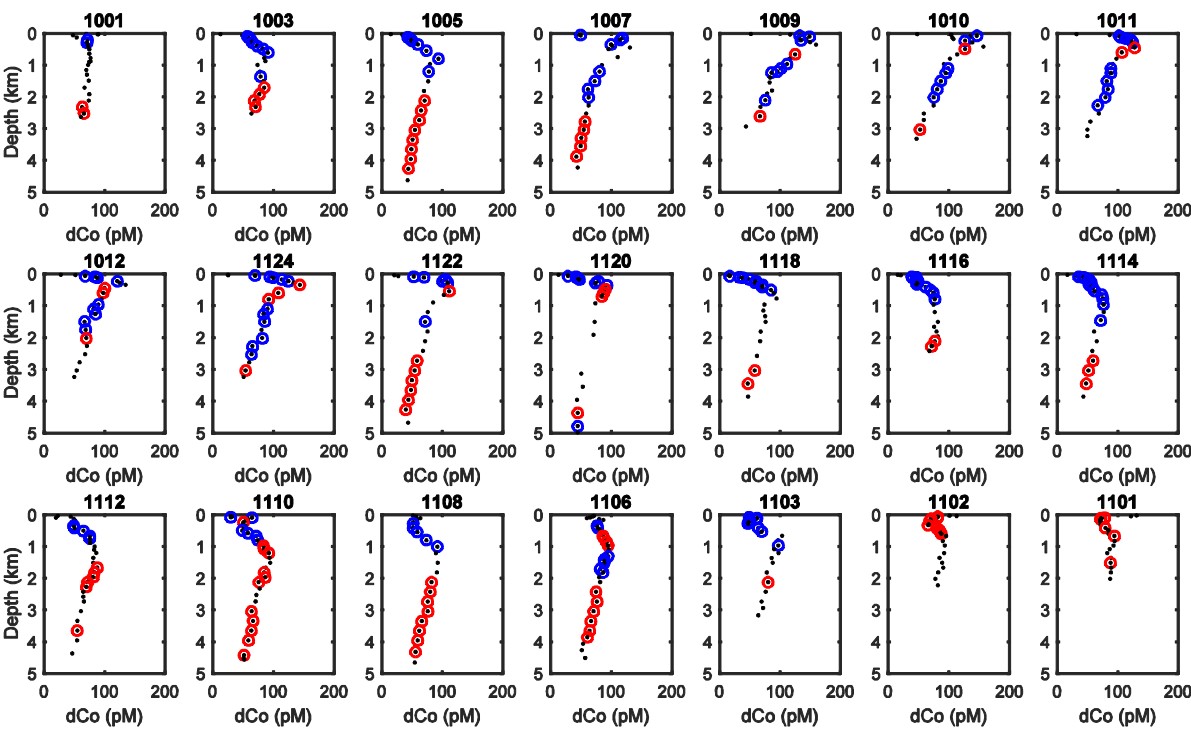



1    Figure 6.

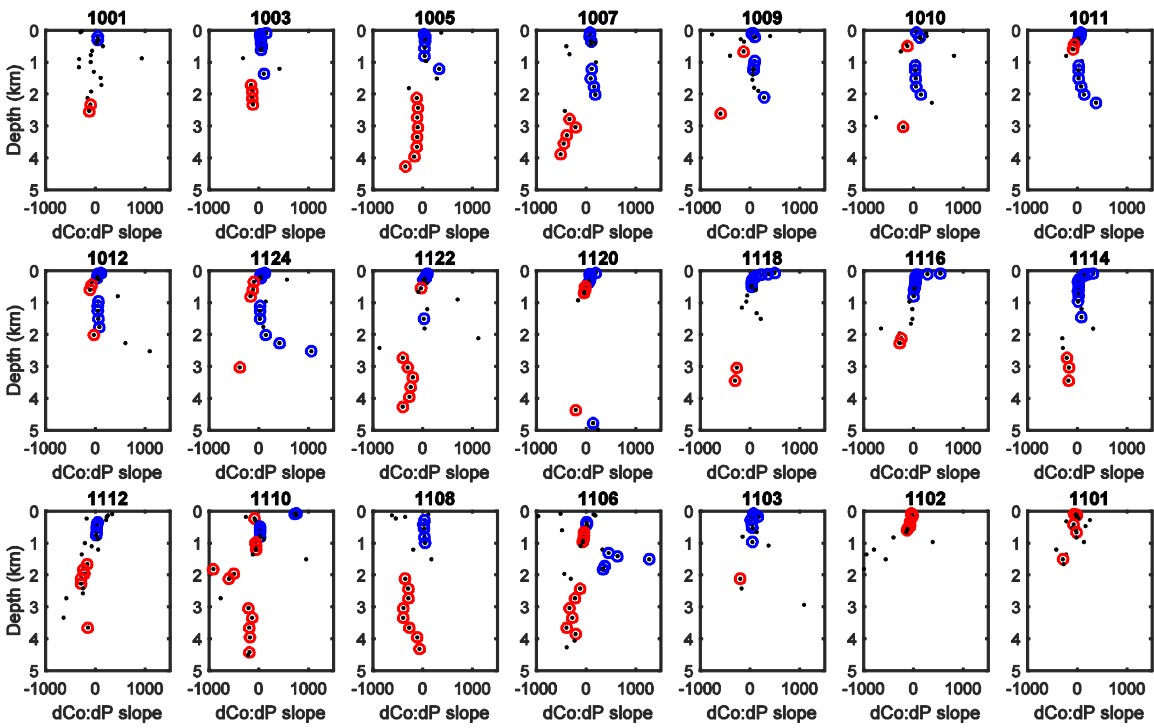



1    Figure 7.

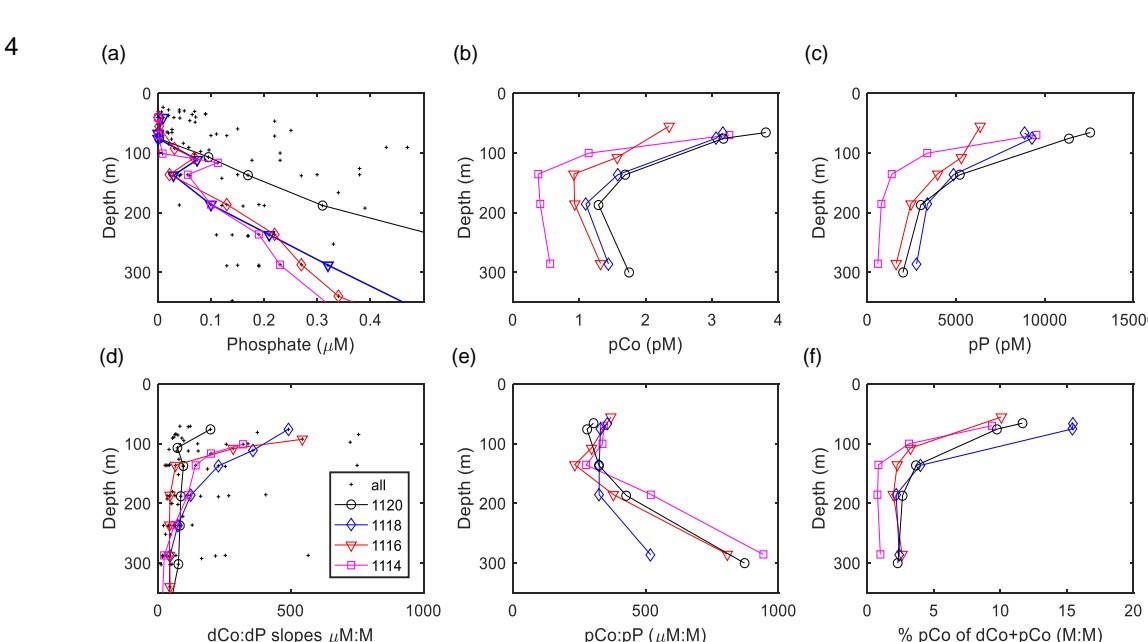





1    Figure 8.

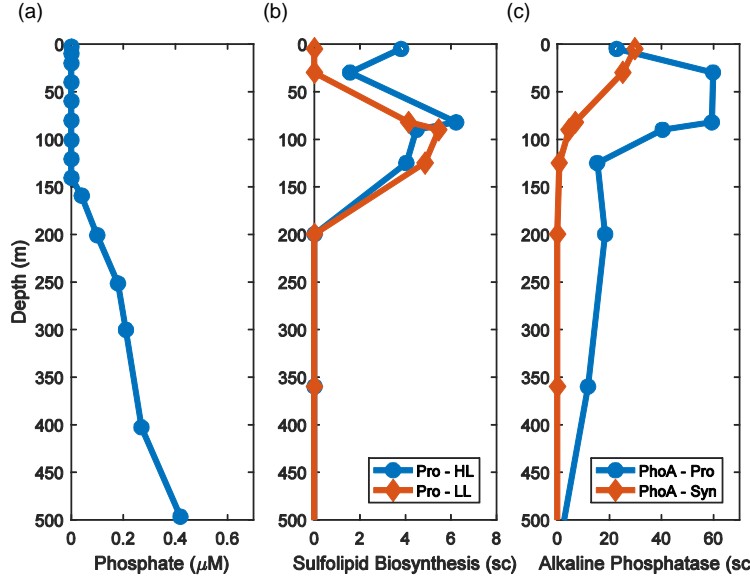





1    Figure 9.

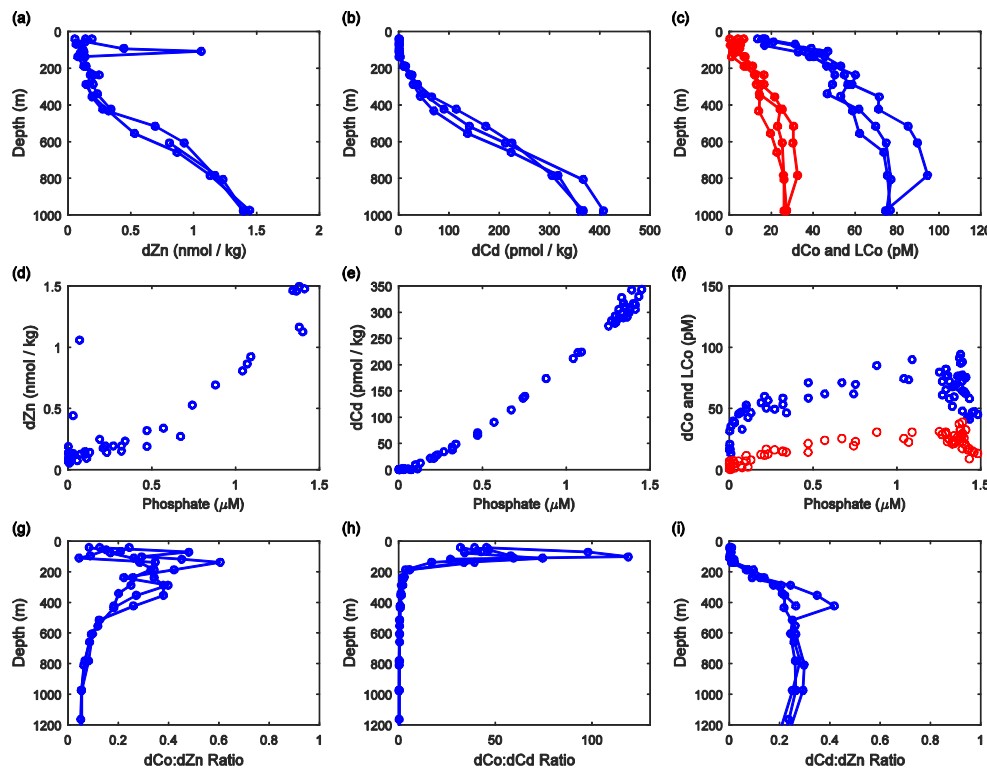



1    Figure 10.

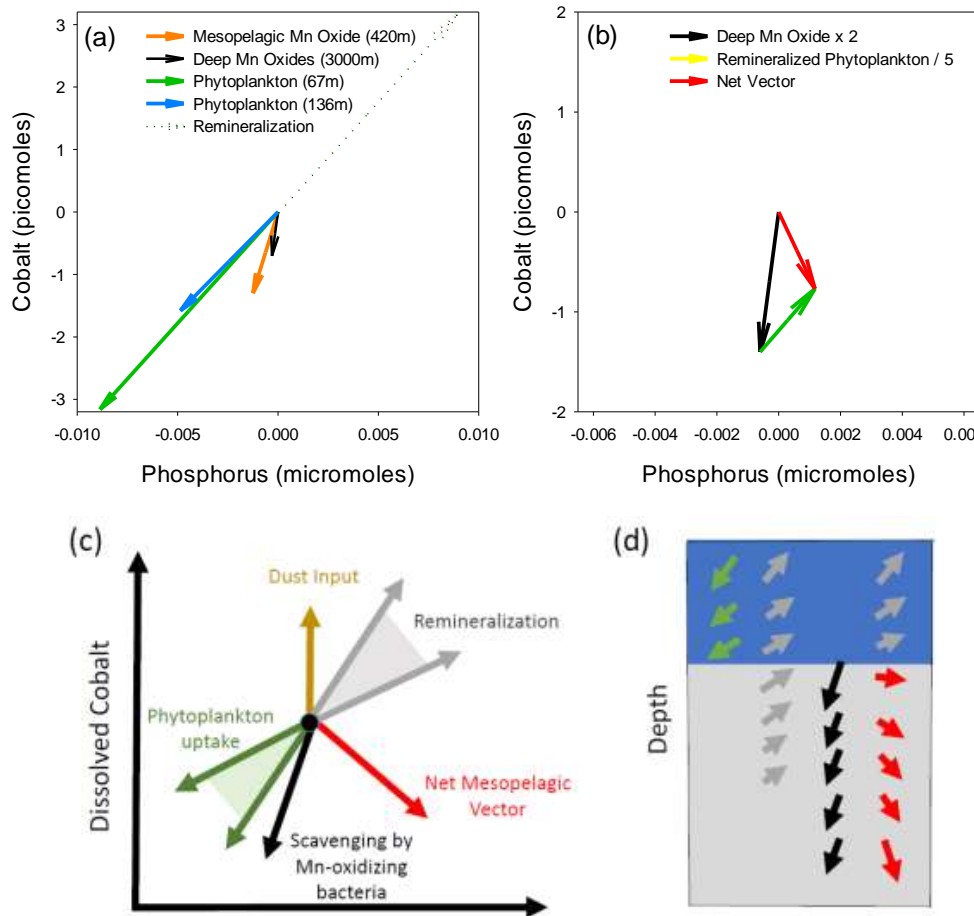





1    Figure 11.

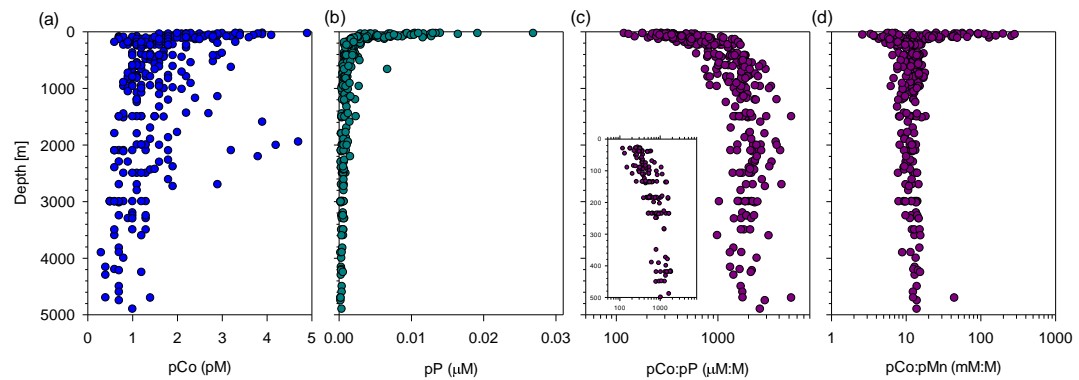





1    Figure 12.

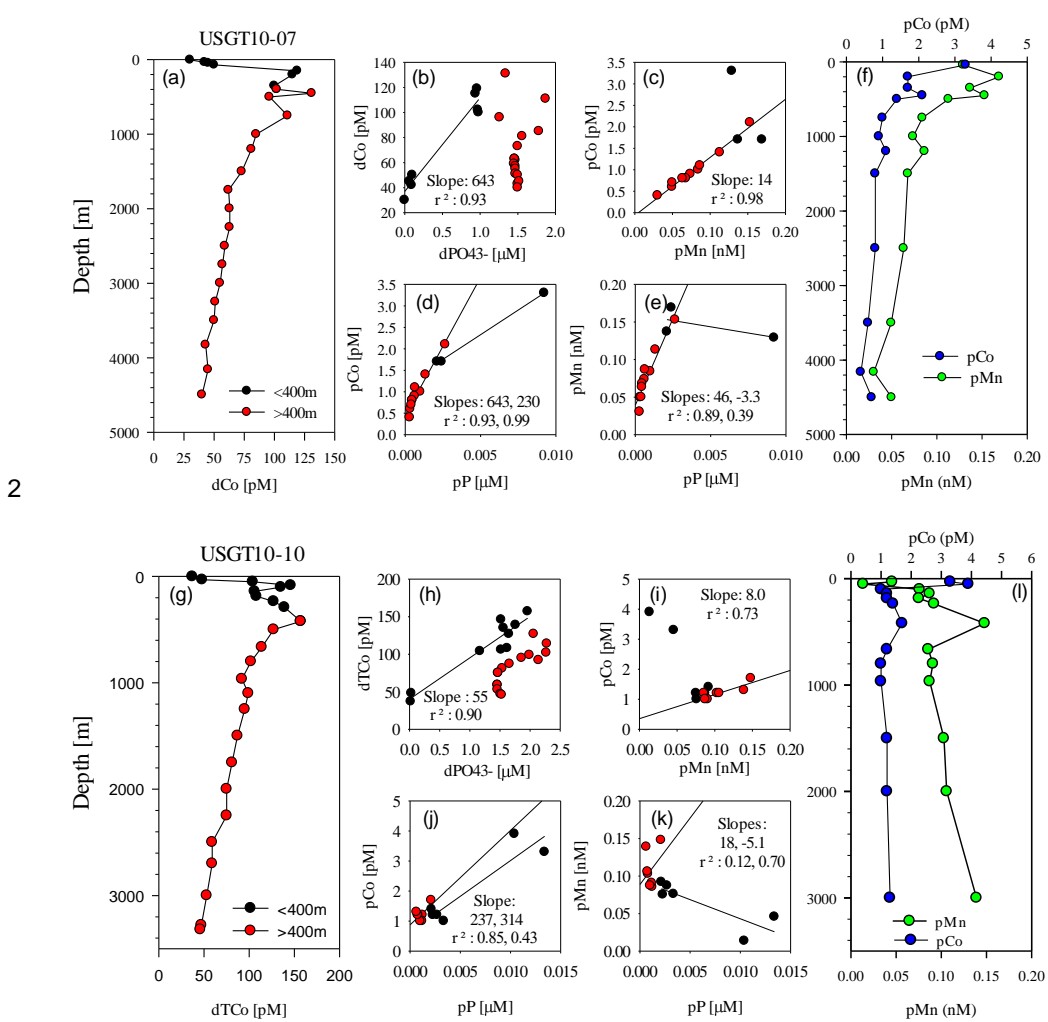





1    Fig. 13.

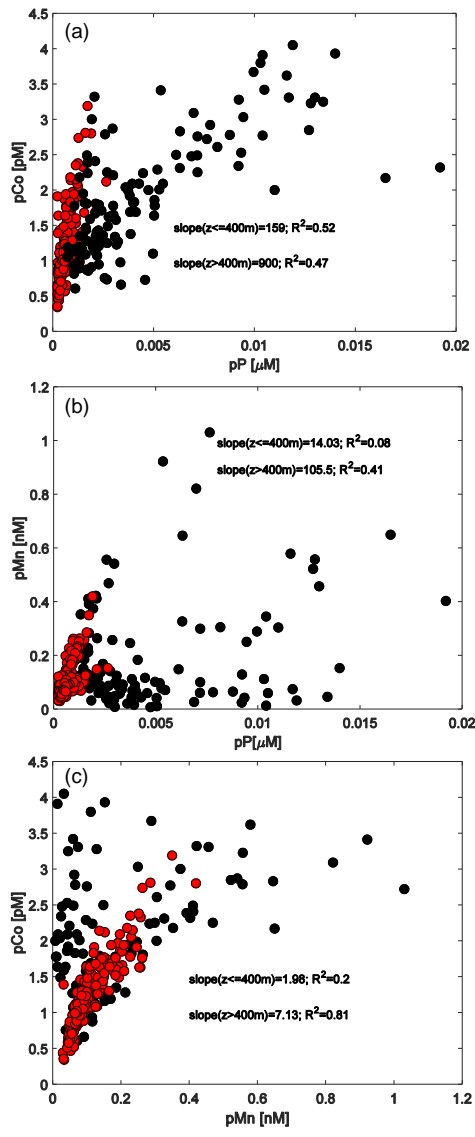



1    Figure 14.

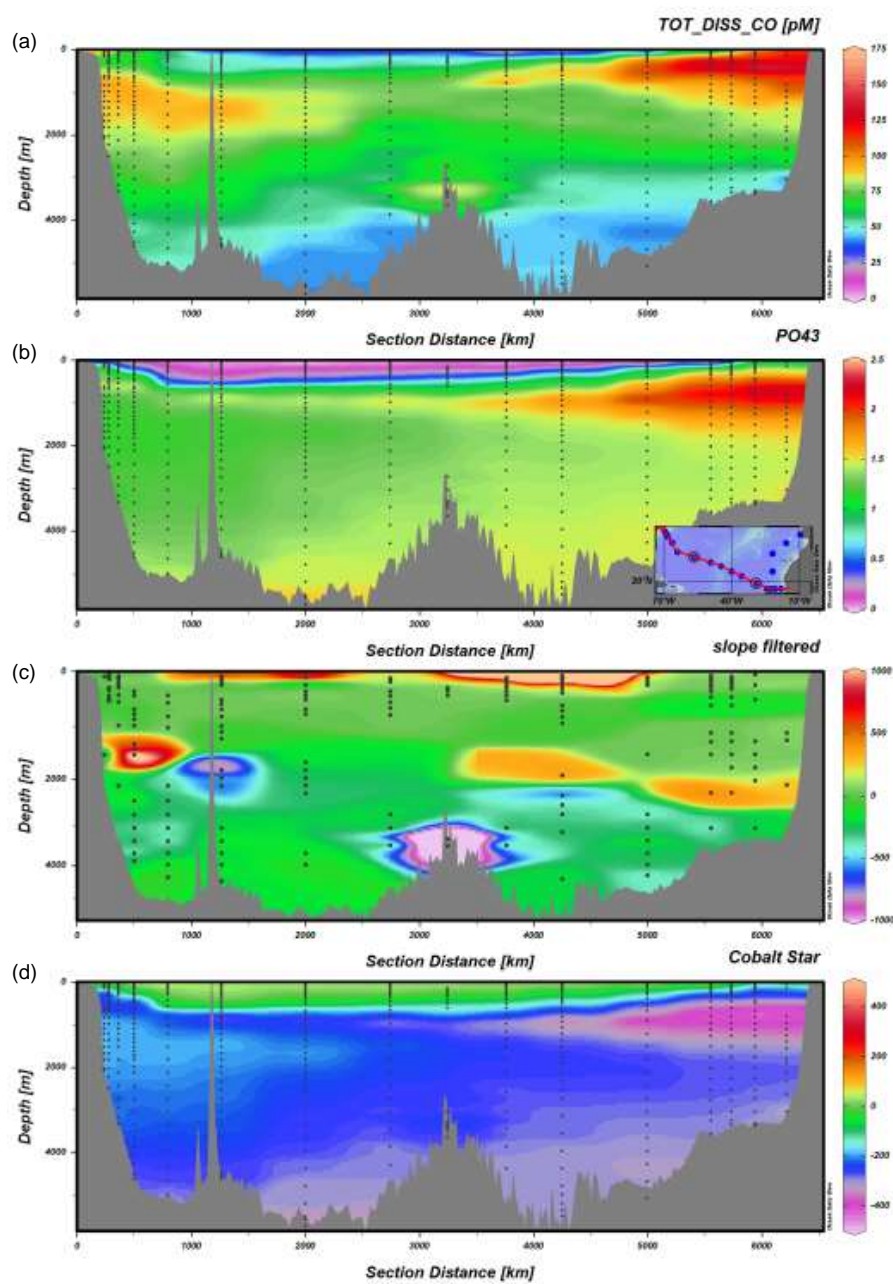



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
