# Peer review of "The Acceleration of Dissolved Cobalt's Ecological Stoichiometry Due to Biological Uptake, Remineralization, and Scavenging in the Atlantic Ocean"

_Biogeosciences, 2016_

## Referee Comment (RC1) · Dr Shelley (Referee) · 18 Jan 2017

Review of bg-2016-511, The Acceleration of Dissolved Cobalt's Ecological Stoichiometry due to Biological Uptake, Remineralization, and Scavenging in the Atlantic Ocean by Saito et al.

General comments I really enjoyed reading this manuscript, which comprehensively describes the large Co dataset from the Saito group's recent Atlantic Ocean expeditions. The issue of Co scavenging is slightly contentious. The sections describing the evidence for Co scavenging based on this data are compelling. I recommend this

manuscript for publication, with only some minor points requiring clarification.

Specific comments P4, line 14: Insert "the determination of" before dissolved cobalt P5, line 5: Corresponding P9, line 42: Occurs P9, line 43: phosphatase P10, line 9: Perhaps "(conserving)" should appear after the first appearance of "sparing" in the Abstract and Introduction P10, line 14: Have P10, line 21, and the corresponding reference: The citation should be for Groß et al. 2013 P10, line 33: "...of the total cobalt..." P11, line 20: Replace "through" with "after" of "following" P11, line 37: Replace "towed" with the approximate depth of the samples, ∼2 m P12, line 4: "...the depth distributions..." P12, line 9: "...protein concentrations and their metal contents..." P12, line 14: ...subtropical gyre..." P12, line 22: You pose the question, "Could zinc have been scarce enough to encourage zinc-cobalt substitution within metalloenzymes such as alkaline phosphatase?", but do not explicitly answer until the end of Section 3.2.6. While this not a problem in itself, you might consider moving the paragraph starting at line 37 to the end of Section 3.2.6, and adding a short sentence to state that you will examine this question in the following two sections (3.2.5 and 3.2.6) P12, line 42: How is "biogeotraces" normally written – BioGEOTRACES? P13, line 4: Add a reference to Figs 2a and b P16, line 30: "vectors" P16, line 37-38: "...dominated by the scavenging removal term (Scav) which transitions..." P18, line 18: Should Ocean Data View be referenced? P18, line 40: "...which was used based on an aggregate..." P20, line 4. This sentence feels out of place here (and at the end of the Abstract). While it points to important future considerations, it doesn't fit with anything else in this manuscript. However, having read the complementary manuscript by Noble et al, I understand where the authors are coming from. I suggest adding a sentence linking this sentence with the more in depth discussion on this subject that can be found in the Noble manuscript. Fig. 2 caption: Define OMPA Fig. 2: The Ross Sea data is not mentioned in the caption Fig. 4a: I cannot read the vector labels Fig. 6 caption: For ease of reading and consistency between this manuscript and Noble et al, you might consider adding a dash to your station numbers, e.g. USGT11-18 or simply 11-18 Fig. 10b: The legend and the vector colours don't match (green and yellow) Fig.

14. How useful a metric is Co* given the challenges in defining Q? Although, it is telling to see negative Co* in the eastern plume

---

## Referee Comment (RC2) · Dr Boye (Referee) · 25 Apr 2017

**Review of "The acceleration of dissolved cobalt's ecological stoichiometry due to biological uptake, remineralization and scavenging in the Atlantic Ocean" (*Saito et al.*) by Gabriel Dulaquais and Marie Boye**

The manuscript provides a new and large dataset of quality of the scarcest micronutrient cobalt. This study brings some new approaches of the interpretation of the cobalt biogeochemical cycle with the connection of rare data of metaproteomic metalloenzyme distribution. This manuscript is generally well written and clearly organized, but it is really long and it could be shortened in several places.

The publication of the manuscript will be obviously of great interest for the scientific community. However, it needs major revisions and clarifications on several points (see below), new calculations and perhaps additional data.

All this approach based on multifactorial calculations of the processes is indeed little convincing without the detail and the methodology of the used calculations. Furthermore, important processes are missing in these attempts of qualitative budget of the cycle of Co and its ratio with phosphate in the Atlantic Ocean. The terms of physical dynamics (as the advection of the bodies of water) are not indeed taken into account, and the external sources (as the atmospheric deposition) poorly considered. It is problematic since it has been previously shown that those later terms could be important and significant in the Co budget, especially the circulation and water-masses mixing in the intermediate and deep oceans that can help to close the budget at a basin scale (see papers of Dulaquais et al. in the Atlantic and the Mediterranean Sea). Furthermore, particulate Co is mainly considered as being biogenic in this work (apart when referring to precipitates with manganese oxides), whereas there are evidences of lithogenic origin of pCo in open ocean, notably in areas receiving Saharan dust inputs such as the ITCZ in the North Atlantic, and from resuspension of margin sediments and advection of such Co-enriched waters.

We thus recommend considering those missing terms and to detail the digital methodology used before being able to publish this paper.

**General comments**

The vector analysis is a very new and interesting approach. Nevertheless, we are concerned at first by the robustness of the approach, since the authors consider a vector significant for $|r| > 0.7$, which means $r^2 > 0.49$. Also by taking into account that the authors use 5 points for the vector regression, the authors should use a value for $r^2$ of 0.77, at least, to consider that the regression is significant. Thus, we recommend to use $|r| > 0.88$ for the interpretation of the vector analysis, or to include 8 points in their regressions. This new criteria for the value of r might change the issues of this work and the interpretation of the data.

In addition, only the middle point is apparently used for the vector analysis, whereas some data before the last depth are considered (e.g.; circled in Figures 5&6), which is not in adequacy with the methodology described in the method part.

Our second major concern in this vector approach is that the authors completely hide the mixing vector. The dilution and mixing processes between the different water masses are indeed not treated in the paper, whereas they could be important in the cycle of Co. Since the authors argue that only scavenging and remineralization govern the deep distribution of cobalt in the deep Atlantic, which fully contrasts with study by Dulaquais et al. (2014a), the authors must include then the mixing vector to confirm their conclusions. An example of the importance of mixing process in the dCo deep distribution is given by its concentrations in NADW and AABW that are, respectively, of ~ 40 pM and ~70-80 pM for dCo, and of 1.25 and 2.25 for soluble reactive phosphates. Thus, mixing of these 2 water-masses gives by a South East vector that appears to be similar to the vector observed by authors. Consideration of these dynamic processes could in turn reinforce their conclusions, or on the contrary completely change them. Another example is the AABW. By following the reasoning of the authors, dCo should decrease in this water-mass from south to north. However instead, dCo increases along its northern pathway in the Atlantic as observed by Dulaquais et al. (2014). This northward increase is also visible in the authors's data set when combining their CoFemug and NAZT cruises datasets.

One possibility to remedy with dynamic processes would be to add advection vector between the different dCo end-members measured in the different water-masses. Authors can also use their datasets from NAZT and CoFeMug cruises, and combine them with the one's of Dulaquais et al. (2014a) in this way. Since the complete analysis of water-masses is available (as performed by Jenkins et al., 2015), the combination of the different end-members weighted by the proportion of each water-mass will help to determine if a located depth is a source (remineralization) or a sink (scavenging) for dCo in the deep sea. Finally, when advection will be included in the vector analyzes and if it is significant, section 3.3 will have to be revisited.

We are also concerned about how authors determined the length of the vectors in their analyzed because the direction of a vector does not translate the intensity of it, thus scavenging might be of strong influence for the direction of the vector but its length is maybe small and poorly influences the distribution. In connection with this, the vector direction strongly depends of the magnitude of each component, thus estimation of Mn oxides and remineralization component will change by adding a third mixing vector.

In their vector approach, the authors do not take into account also the possibility of a differential remineralization between Co and P (in section 3.1, p 7, lines 1-11). Differential remineralization between macro- and micro-nutrients is now admitted and such differential remineralization seems to occur in these waters, as evidenced by the negative Co* values obtained in the Eastern Atlantic OMZ. In OMZs, oxidative scavenging of dCo is indeed considerably limited (Hawco et al., 2016; Noble et al., 2012), thus if Co* is negative in this remineralization zone, it probably indicates that Co is less remineralized than P.

Dulaquais et al. studies in the Atlantic (2014a, 2014b) are poorly cited for their observations and measurements of dCo along the pathway of the different water masses, and they are even occulted for the dCo/P correlations they recorded in the Sargasso Sea and the Equatorial Atlantic, which are nevertheless close to those reported in this study. These works should be included across the different parts of the manuscript when it is pertinent (remineralization, Co-P correlation, possible transport of Co across the different water masses).

In addition, Co* plot seems to be in accordance with Dulaquais et al. studies: Co* is positive in the surface layer that could be interpreted by a higher biological remobilization of Co than P in the upper layer (Dulaquais et al., 2014b), and its negative value in the remineralization zone of the OMZ can be interpreted by a lower remineralization of Co than P (Dulaquais et al., 2014a).

We are concerned about uncertainties on the phosphate measurements in the North Atlantic subtropical gyre (in section 3.2.3), wherein these phosphate concentrations are the lowest recorded in the surface ocean. Uncertainties on the slopes and correlations recorded should be noticed in regards with phosphate analyses.

The idea of a one-way back from dissolved to particulate fraction in the mesopelagic zone (section 3.2.7, p 15, lines 7-15) can be more robust by using particulate labile Co data in the deep sea, which might be available from Twining et al. (2015). If not available, this consideration should be weighted because it is in contradiction with Moffett and Ho (1996) study "In the Sargasso Sea […] cobalt uptake was nonoxidative, biologically mediated, and enhanced by low to moderate levels of light. It is probably due primarily to uptake by phytoplankton". Moreover, Wu et al. (2014) estimated that the oxydation of Mn is restricted in the deep Atlantic along the GA03 section, thus co-oxydation of Co should also be restricted.

The correlation between pCo and pMn is indicative of a covariance between the two elements in the particulate fraction (section 3.3, p 17-18), but it does not indicate that the vertical particulate flux (and thus scavenging) of cobalt increases with depth. Sediment traps or $^{234}$Th data are indeed needed to constrain this flux. Alternatively, this flux can be modeled at a first glance using Martin's curves and Co/C data set probably available from this cruise.

The strong Co-Mn correlation can be interpreted in another way than that actually proposed: it is indeed possible that the biogenic Co fraction is remineralized while the authigenic fraction (scavenged fraction) is not (i.e.; the one way trip of Co presented by authors). In these conditions, even if scavenging is slow and restricted, particulate Co partitioning will shift from the biogenic (organic) to the authigenic (mineral) fraction and a strong correlation between Mn and Co will appear. This possibility can be further assessed using the data presented in this study: for instance the particulate Co:P correlation gives a slope of 159 µM:M in the upper 400 m and shifts to a constant 900 µM:M in the deep ocean. If cobalt is only scavenged when phosphate is remineralized, the Co:P slope should gradually shift to a negative value, given a high pCo (scavenging) when pP is low (remineralization). Because it is not observed, it is possible then that the biogenic pCo fraction is decreasing with depth while authigenic is increasing with depth. Then, the particulate Co:P ratio can be parameterized as following:

Mean Co:P = $x$ * (Co:P)$_{biogenic}$ + $y$*(Co:P)$_{authigenic}$ with $x$ and $y$ [0;1] and $x + y = 1$, and (Co:P)$_{biogenic}$ << (Co:P)$_{authigenic}$

In these conditions, the Co:P slope will gradually increase with depth to a maximum when all the biogenic P is remineralized, as it is observed by authors (Figure 11c). Adding a z color scale for depth on pCo:pP data (in Figure 13) may also help to discern such shift. Moreover, if Co is less efficiently remineralized than P, this Co:P will also be enhanced.

The occurrence of scavenging is highly probable in the deep sea (and it is not the debate here) but because it timescale is slow (Noble et al., 2012), estimating its magnitude is crucial to compare it with other fluxes (advection) and to determine which terms govern the distribution of dCo in the deep Atlantic. Scavenging can occur at a low rate and thus can less impact the distribution than mixing processes. To firmly demonstrate that scavenging is only driving the deep distribution of dCo, additional data of particles fluxes are needed. These estimations of fluxes will serve to quantify the scavenging flux and rate, and then, to compare with the rate of mixing at each layer. Then, if the scavenging rate overwhelms mixing it will be demonstrated, otherwise an alternative vision has to be included in the discussion.

The Q used for cobalt star does not seem pertinent regarding the value recorded by Twining et al. (2015) of 150 µM:M. The Q used by authors must be the one recorded in phytoplankton especially if caveats exist on the soluble reactive phosphate measurements at extremely low concentrations. Negative Co* observed in the OMZ should be discussed. If a major source of dCo occurs in this area, Co* should be positive and not negative.

Conclusion would probably have to be modified after inclusion of new calculations and mixing estimations. The lithium ion batteries comment, which is never discussed in the paper, should be removed.

**Specific comments**

Abstract: The lithium ion battery comment in the abstract is not discussed elsewhere in the text, it has to be removed then.

P 4: The method part for determination of dCo is poorly documented in the manuscript. Even if the method is detailed in the accompanying manuscript, the limit of the detection of the method, the reference water measurements and the relative standard errors are specifically needed because of the use of the vector analysis.

P 5: Since surface phosphate concentrations are extremely low in the subtropical Atlantic and because phosphate data are crucial for the analyses of the acceleration of the dCo/P ratio and the vector analysis, the limit of detection and quantification of soluble reactive phosphate (RSP) should explicitly be indicated in the method part, as well as the lowest concentration of RSP measured in this study for the interpretation of the data. Considering the extremely low SRP measured, caveats on the dCo/P correlation at low phosphate concentrations must be mentioned.

P 7: The high dCo:P value observed in the north Atlantic subtropical gyre (67 µM:M) is in excellent agreement with Dulaquais et al. (2014b) of 64 µM:M, and it should be noted.

P 8, lines 1-4 ("Labile cobalt is likely highly bioavailable relative to complexed cobalt"): According to the method, labile cobalt is also complexed and is not equivalent to inorganic Co. It is the fraction that can exchange with the added ligand at the potential of deposition. This fraction is dependent of several factors and cannot be defined as the bioavailable fraction without evidences. Furthermore, in previous studies of the authors, complexed cobalt is described as the bioavailable fraction for cyanobacteria that produce strong organic ligands (Saito et al., 2003; Saito et al., 2005). These two opposite views should be better explained here.

Section 3.2.6: Authors used a C/P ratio of 106:1, which is 2-3 times lower than the ratio given by Bertilson et al. (2003) in depleted cultures, and 1.5 times lower than the one's recorded by Twining et al. (2015) in picoplankton (of 153.1 ± 26). We recommend using the values recorded in phytoplankton cell during the cruises.

P 15, line 20: Moffett and Ho have observed the prevalence of biological uptake in Sargasso seawater, and that microbial oxidation was predominant in estuarine seawater.

P 16, lines 1-7: It will be demonstrated only when the advection vector will be implemented. If the advection vector is in the same direction, scavenging, remineralization and advection will have to be quantified to estimate the length of the vectors.

P16 (equation 1): Error in the denominator (should be dz). Advection must be included.

Caption of Figures 3 and 4: may have troubles in labeling the sub-figures.

Table 1: Dulaquais et al. (2014b) also recorded Co/P correlation in the Atlantic and it has to be integrated.

Figure 2: The total dissolved Co axes should be on the same scales. What about the SACW? Why the Ross Sea is included in the figures?

Figure 3: Error in the caption (should be (b), (f); instead of (b), (e)). Only show vectors when r > 0.88

Figure 4: is not easy readable. Inserts must be bigger and include mixing vector in Figure 4a. Error in the caption (should be "blue", instead of "black", when referring to positive slopes). In Figure 4a&d, depth can be added as a z parameter. The sum of the positive and negative vector weighted to the intensity of the dCo/P correlation might be useful to see whether positive or negative slopes exceed vector. Figure 4 b&e are not really used for the discussion.

Figure 5: Some bottom depths and depth just above them are circled and it is not in accordance with the 5 points regression method used.

Figure 6: Uncertainties on the vector should be added, and only vector with $r^2 > 0.77$ should be presented.

Figure 7: dCo data are needed here.

Figure 10: Mixing vector must be included.

Figure 12e-j-k: Are correlations with a $r^2$ of 0.39, 0.43, 0.12; and calculated on 3-6 points really significant?

Figure 13: can add color scale as z axis.

Figure 14: Isoclines are needed and focus on the upper 1000 meters also.

---

## Author Comment (AC1) · 22 May 2017

We thank the reviewer(s) for their comments on the manuscript. We provide point-by-point responses here, although we were not able to obtain a digital (non-pdf) version of the review so reviewer comments are paraphrased or quoted. In general, we feel that we can incorporate the reviewer's suggestions or in some cases there was a misunderstanding about the data or interpretation they are interested in being elsewhere in the paper. I think the major scientific point of this paper is that dissolved Co : phosphate ratios are extraordinarily diverse, far more so than for any other macro or micronutrient

by a large margin. This paper attempts to capture the full extent of this stoichiometric diversity particularly in its effect on the dissolved phase, with novel connections to the particulate and biochemical phases. To do so required employing some new approaches using simple statistical methods, and using the power and confidence associated with a large number of statistical analyses rather than stringent methods on single or few analyses of a large dataset as done previously.

The reviewers ask about the methods used: "all this approach based on multifactorial calculations of the processes is indeed little convincing without the detail and the methodology of the used calculations". There are two issues here. First, regarding the methods, there is no multifactorial analysis, simply a Matlab script that repeats a simple two-way linear regression many times on groupings of 5 depths many times across the sections in order to achieve the high-resolution stoichiometry targeted in this study. This method is simple and is fully documented in the methods. We will add suggested additional information about the length of the vectors. The vector addition is also documented in the paper already in the discussion, and is used primarily as an example scenario (e.g. what vectors could be added to result in the observed negative vector).

Second, regarding the question about the linear regression approach - we understand the reviewer's concern about statistical choices. Specifically, the reviewer suggests increasing the threshold from r= |0.7| to |0.88| and the number of data points per regression from 5 to 8. In response, we point out that this exact broader scale analysis with more data and higher r2 values was conducted and presented in this manuscript as described in Section 3.2 as "aggregate" datasets (and the regressions figures themselves are shown in Noble et al., 2017 BG in press Figure 11, r2 vary from 0.67 to 0.92). After presenting these large scale and more significant features, we argue that there is large amount of upper euphotic zone data that was excluded in order to acquire these high r2 regressions, yet likely represents real increased stoichiometric features. This "profile-based" analysis approach was applied to measure these shallow phenomena. The high resolution approach requires smaller depth intervals in order to capture the resolution because they are increasing (and accelerating) towards the surface. For many years we have observed a significant increase in slope in the upper photic zone which we have believed to be a real biochemical effect. With sectional datasets now available, we are able to combine dissolved, particulate, and biochemical protein datasets in this manuscript to study this phenomenon. For the dissolved samples to be expanded to 8 depth would result in a loss of depth resolution in which these features would be obscured (this effectively imposes an assumption of a constant stoichiometry rather than an increasing one). We can present all of the r value data that is above our threshold so the reader can evaluate the quality of the regressions themselves (this used to be part of Fig 3 but we removed it to make the figure panels larger). To change to 8 points and arbitrarily higher r values thresholds is ignoring the motivation for this study to tease a subtle but apparently widespread signal from the euphotic zone, which is also supported by particulate and protein data here and which is found across two ocean sections.

The reviewer also comments on how: "Only the middle point is used in vector analysis". This is a misunderstanding. For each 5-depth point regression the resulting slopes are plotted centered around the data, which is the midpoint within the 5-points. We could easily switch the choice of depth to be any of the 5-depth points within the regression, but centering at its midpoint seemed most logical.

The reviewers' second main concern is regarding the lack of consideration of a mixing vector in the vector analysis. The reviewers incorrectly state that we "argue that only remineralization and scavenging govern the deep distribution of cobalt in the North Atlantic". On the contrary we state that "These scavenging signals co-occur with distinct water masses identified by OMPA analysis, implying that these scavenging processes are being integrated on decadal-to-century timescales of deepwater circulation processes within the ocean interior." While it is true that water mass mixing could alter the ratio of Co and P, mixing itself is not a process that removes or add Co or P from the dissolved phase (except in estuaries). Moreover, the ratios of the dCo and PO4

provided in the review produce an almost identical ratio of 32 and 31-35 (40, 70-80pM dCo; 1.25 and 2.25 PO4), so mixing these NADW and AABW water masses would not create a significant vector in Co:P space by their example. We could add a mixing vector to the plot as we did in our original vector diagram for upwelling (Noble et al., 2008). Our point here is that in these localized fine scale regression analyses vertical processes of uptake remineralization and scavenging, whose signals are clearly integrated through horizontal advection, can explain the large variation and change in sign of the Co:P slopes. For a revision we can add in a mixing vector and/or further elaborate and clarify on the role of mixing in vector analyses.

The reviewer suggests using the water mass analysis in this study of end member distributions: "since the complete analysis of water-masses is available, as performed by Jenkins et al., 2015), the combination of end-members weighted by the proportion of each water-mass will help to determine if a located depth is a source (remineralization) or a sink (scavenging) for dCo in the deep sea." We point out that this water analysis has already been included and is shown in Figure 2 for the North and South Atlantic Zonal sections including the following text that relates to the reviewer's suggestion. "These scavenging signals co-occur with distinct water masses identified by OMPA analysis, implying that these scavenging processes are being integrated on decadal-to-century timescales of deepwater circulation processes within the ocean interior (Noble et al., submitted). Specifically, negative slopes water masses were found to be in the Denmark Straits Overflow Water/Antarctic Bottom Water/Iceland Scotland Overflow Water (DSOW/AABW/ISOW) and Classical Labrador Seawater (CLSW; Fig. 2) water masses both of which have long deepwater transit times (Jenkins et al., 2015)." We argue that the negative vectors we observe are the influence of slow scavenging processes accumulating on isopyncals during advection, and hence that the influence of scavenging and mixing are combined in the negative vectors we observe. The reviewers imply that mixing of Co is a conservative process (without any scavenging), but we respectively disagree with this argument from Dulaquais et al., 2014, and argue that the data shown here, and in a more substantive manuscript on Co scavenging (Hawco

et al, Marine Chemistry in revision) provide evidence of mesopelagic scavenging. In particular, in the latter manuscript a comparison of dCo with C14 ages in DIC showed basin scale scavenging processes between the Atlantic and Pacific basins. Much of the reviewer's interest in scavenging is more pertinent to the Hawco et al. manuscript, while the present manuscript focuses on the dynamic variation in dCo:P relationships in the vertical dimension, which must include some discussion on the unusual negative slopes in the mesopelagic and deep ocean and how the shift from remineralization to scavenging can result in the transition in positive to negative slopes as shown in Figure 10. The reviewer states that we are inconsistent with Moffett and Ho's Co and Mn microbial oxidation dataset which did not find measurable Mn oxidation at BATS. This is incorrect, Moffett and Ho found phytoplankton uptake dominates within the euphotic zone at 60m, which is consistent with our uptake/remineralization vector based on positive slopes at this depth. Moffett and Ho did not present uptake rates in the mesopelagic in that manuscript.

The reviewer suggests an alternate means for particulate Co-Mn correlations, "The strong Co–‐Mn correlation can be interpreted in another way than that actually proposed: it is indeed possible that the biogenic Co fraction is remineralized while the authigenic fraction (scavenged fraction) is not (i.e.; the one way trip of Co presented by authors)". We appreciate the reviewers considering and describing this scenario in detail. This interpretation is actually the phenomenon we are presenting and arguing for in every aspect they discuss (an increasing importance of scavenged/authigenic particulate for pCo and pMn with depth resulting in the pCo:pMn relationship and the increased pCo:pP relationship). We are glad to hear they agree! We point out that it is this authigenic pCo formation process that in turn creates the negative dCo:dPO4 slopes in the dissolved phase we have been debating above. We have found that the connections between dissolved and particulate phases are fascinating yet can also be confusing and take time to fully consider given the mirroring/opposite effects the phases can have on each other and out community's limited experience in direct comparisons of dissolved and particulate data.

The reviewer states that the vector approach does not take into account differential remineralization. This is an interesting point, but I think it is not true. The profile-based regression analysis does not include any a priori assumption about stoichiometry or remineralization efficiency. It is simply a measurement of dCo and dP change in response to each other in the vertical dimension. There were no Co* values reported in Hawco 2016 or Noble 2012, but it is here. I understand the hypothesis: an alternate explanation for vertical structure is the preferential remineralization of P over Co creating the Co deficiency, as an alternative to scavenging. This is a useful alternate perspective that we will add to the discussion. Reflecting on it now, there are several datasets that argue against differential remineralization being the major driver in dCo distributions: 1) the scavenged like profile is inconsistent with this (implying removal rather than slowed accumulation), 2) the correlation of pCo and pMn in the mesopelagic is consistent with biotic Mn oxidation rather than with slow Co remineralization (which would not create a correlation with Mn), and 3) the negative slopes of dCo:P are consistent with Co loss rather than slowed accumulation (which would be a positive slope still). To clarify the prior and current observation of increased dCo in the OMZ plumes were not interpreted as a result of slowed / differential remineralization but reduced Mn and Co scavenging.

The reviewer suggests incorporating data from Dulaquais et al into the study. There is already a lot of data within this study and there were some intercalibration challenges at the western portion of the North Atlantic. As a result, we think this approach investigating the large diversity of Co:P relationships is useful as a beginning, and we encourage those authors to examine their datasets with these methods in the future.

The reviewer comments on our choice of a Co* ratio from the pCo:pP North Atlantic ratio, and argues that it should instead use the lower Twining from phytoplankton species. This is a good comment and one we also wrestled with. We point out that Redfield et al's original studies used oceanic particulate carbon, nitrogen and phosphate ratios as well, and they have been interpreted as being reflective of the aggregate community, so

there is precedent for working from the aggregate particulate phase. The cell specific numbers from Twining et al. are taxon-specific and span 140-3000 depending on the group (their Table 5). We were transparent about the challenge of selecting a single Co:P in the manuscript, yet we feel the cohesive structure in the visualization of Co* is compelling, even if its interpretation is not fully understood as of yet. We could change our value to the suggested value, and will consider this in the revision. We point out that though that the choice of any the Co:P value will not affect the trends observed in this visualization, since any change in Co:P will be applied linearly to all Co* values.

Specific comment section:

Regarding the statement about growing human use of cobalt: we can remove these statements. They are expanded and referenced briefly in the accompanying BG manuscript. Although statements of context and relevance seem reasonable, for example papers studying mercury biogeochemistry include a sentence or two stating its environmental relevance. Since most people are unaware that lithium ion batteries are in most cases lithium cobalt batteries, raising awareness of their connection to ocean biogeochemistry seems a worthy goal in raising public awareness.

The documentation for the method for dCo is presented in the accompanying in press manuscript Noble et al., BG. This is a data analysis paper, rather than a data presentation paper. We can duplicate it here perhaps in abbreviated form though.

We have requested detection limit data for the phosphate measurements, and will add them to the methods as requested. For many of the shallowest samples in GA03 the nutrient team did not provide values (presumably below DL values) and hence there is no correlation given.

We will comment on agreement of Co:P with Dulaquais et al., thank you for pointing this out.

The reviewer asks for clarification about the bioavailability of labile cobalt and complexed cobalt. They state: "according to the method labile cobalt is also complexed and is not equivalent to inorganic cobalt". We disagree with this somewhat – the labile cobalt may be inorganic or weakly bound cobalt. The reviewer states that we have previously stated that complexed cobalt is bioavailable and asked us to clarify this. We have argued and presented data to show that some phytoplankton, namely the cyanobacteria, can access strongly complexed cobalt. This seems reasonable since they have an absolute requirement for Co (albeit a very small requirement) and they live in the euphotic zone where there is typically no detectable labile cobalt. Cultures studies of eukaryotic phytoplankton have been consistent with their use of only inorganic cobalt thus far.

The reviewer suggests using the a P-limited C:P ratio for our conversion of Sunda and Huntsman's Co:P and Zn:P dataset (1995). We point out that that study was not P-limited so that is not appropriate here. This would be appropriate for our environmental datasets, but fortunately we are using Co:P data rather than Co:C data so no conversion is needed.

Fig 12 low r2 values: we are not arguing these are significant, but showing the trends obviously have a negative rather than positive slope. Would could remove the r2 values, but wanted to be fully transparent since we are showing the negative slope lines. We can either remove the lines or state that we do not imply they are significant other than having a negative sign.

Fig 6: Slopes reported on bottom values. That is a good point, we will double check the script as to why it is being reported.

The reviewers state: "p 16 lines 1-7: It will be demonstrated only when the advection vector will be implemented. If the advection vector is the same direction, scavenging, remineralization and advection will have to be quantified to estimate the length of the vectors." As discussed above, our interpretation of the scavenging vector is that it is a gradual accumulation of scavenging of Co upon the water mass during circulation, and

hence advection is included. We can further clarify this in the text.

The reviewer comments on how Co is being considered largely from a biogenic and scavenged phase. We can include a lithogenic correction in a subsequent version for the particulate phase.

Other small comments are repeats of from above or will be corrected/attended to.

We appreciate the reviewer's efforts in this extensive review.

---

## Author Comment (AC2) · 22 May 2017

We thank the reviewer for her efforts. We will make/consider all of the requested changes or suggestions.

Regarding the use of Co*, we agree that it is a challenging value to estimate with a highly variable Co quota value. Yet as the reviewer points out it is perhaps more useful in comparison to other * variables in having such deficient values in the mesopelagic and deep (including in OMZs where the source is high), demonstrating a global deep ocean deficiency of this metal relative to P. For that reason it may be a useful parameter

to explore and present here, particularly since the visual is rather striking in being cohesive and showing major patterns.

---

## Author Comment (AC3) · 30 May 2017

We thank the reviewers for their efforts which will certainly improve the manuscript. Point-by-point responses have been posted as comments to each review. In particular, we emphasize that we have conducted a two-tiered analysis of the datasets, first in large aggregate analysis of Co:P relationships with large groups of data (and high r2 values). This was followed by many 5-point profile based analyses that was inspired by the observation that both above and below the depth region studied for the aggregate Co:P relationships, there appeared to be important features of Co:P with distinct

stoichiometries. These dissolved relationships were supported by particulate Co and P data, as well as biochemical proteomic analyses to document an acceleration of Co stoichiometry resulting from ecological and biogeochemical changes in the upper euphotic zone. We also point out that many of the concerns of reviewer #2 regarding dust and scavenging are beyond the scope of this study, and have been addressed by us in greater detail in the accompanying manuscript bg-2016-512 (Noble et al., 2017 BG in press) and in a Marine Chemistry manuscript that is now under minor revision (Hawco et al., 2017).

---

## Author Response (AR1)

**Point-by-Point Responses and Accompanying Revisions to BG-2016-511**

We thank both reviewers for their useful comments. We have considered all of their comments and incorporated a number of changes into the manuscript as part of these minor revisions, which has improved it considerably. We provide point-by-point responses below.

**Reviewer #1:** All suggested comments from Dr. Shelley were incorporated, we thank her for her careful reading and suggestions. Because the upper water column has such a small inventory of dissolved cobalt, we felt a brief mention regarding the massive increase in use of lithium-cobalt batteries and cobalt mining near the South Atlantic region would be particularly relevant to this manuscript, and we have added two sentences to the introduction based on this, connecting the potential impact the small inertia of the upper water column cobalt inventory.

**Reviewer #2:** We thank Dr. Boye for her comments on the manuscript. We provide point-by-point responses here, although we were not able to obtain a digital (non-pdf) version of the review so reviewer comments are paraphrased or quoted. In general, we have incorporated the reviewer's suggestions or in some cases there was a misunderstanding about the data or interpretation they are interested in being elsewhere in the paper. I think the major scientific point of this paper is that dissolved Co : phosphate ratios are extraordinarily diverse, far more so than for any other macro or micronutrient by a large margin. This paper attempts to capture the full extent of this stoichiometric diversity particularly in its effect on the dissolved phase, with novel connections to the particulate and biochemical phases. To do so required employing some new approaches using simple statistical methods, and using the power and confidence associated with a large number of statistical analyses rather than stringent methods on single or few analyses of a large dataset as done previously. This focus was highlighted further by some additional sentences in the introduction and altered sentences in the abstract.

The reviewers ask about the methods used: "all this approach based on multifactorial calculations of the processes is indeed little convincing without the detail and the methodology of the used calculations". There are three minor issues here. First, regarding the methods, there is no multifactorial analysis, simply a Matlab script that repeats a simple two-way linear regression many times on groupings of 5 depths many times across the sections in order to achieve the high-resolution stoichiometry targeted in this study. This method is simple and is fully documented in the methods. It could be conducted in Excel or similar software for each 5 data-point analysis, the script serves to iterate the analysis many times over.

Second, the reviewer asked for additional information about the length of the vectors. The length of vectors shown in Figure 3 were purposefully made to be larger than the plot since they do not connote information in this context, and this is stated now in the caption. The vector addition used in Figure 10 is documented in the paper already in the discussion, and is used primarily as an example scenario (e.g. what vectors could be added to result in the observed negative vector). We agree with the reviewer that this could become a useful quantitative analysis in the future, but a detailed characterization of vector length and their meanings is likely beyond the scope of this manuscript.

Third, regarding the question about the linear regression approach, the reviewer suggests increasing the threshold from r= |0.7| to |0.88| and the number of data points per regression from 5 to 8. In response, we point out that this exact broader scale analysis with more data and higher $R^2$ values was conducted

and presented in this manuscript as described in Section 3.2 as "aggregate" datasets (and the regressions figures themselves are shown in Noble et al., 2017 BG, their Figure 11, r2 varying from 0.67 to 0.92). After presenting these large scale and more significant features, we argue that there is a large amount of upper euphotic zone data that was excluded in order to acquire these high r2 regressions, yet likely represents real elevated stoichiometric features. This "profile-based" analysis approach was applied to measure these shallow phenomena that we have long observed, but that simple aggregate linear regressions have been unable to capture effectively. We point out that the reviewer's recent paper also observed the same phenomenon, where in the Dulaquais et al., 2014 dataset for the North Atlantic Gyre, their *suggested value of r2 of greater than 0.77 would exclude their own Co:P value for this region* (their figure 8, R2= 0.65 for n=32). Visually the acceleration of Co:P stoichiometries in the upper photic zone is also apparent in their dataset as well (their Figure 8 again). As a result, we agree with their suggestion that this finer approach to characterizing stoichiometries could be useful on multiple section analyses (but decline to include more data in this already large manuscript).

This demonstrates the motivation for conducting the moving depth window regression approach in our study, in that trying to fit a single line to a stoichiometry that is accelerating creates errors and hence we argue that our combination of aggregate dataset and moving window regressions is able to capture the bulk and fine-scale features within the dynamic Co:P ratios in the upper water column. The high resolution approach requires smaller depth intervals in order to capture the resolution because they are increasing (and accelerating) towards the surface. For many years we have observed a significant increase in slope in the upper photic zone which we have believed to be a real biochemical effect. With sectional datasets now available, we are able to combine dissolved, particulate, and biochemical protein datasets in this manuscript to study this phenomenon. For the dissolved samples to be expanded to 8 depths from 5 would result in a loss of depth resolution in which these features would be obscured (this effectively imposes an assumption of a constant stoichiometry rather than an increasing one). We have altered our Figure 3 to present all of the r value data that is above our threshold to be fully transparent to the reader. To change to 8 points and arbitrarily higher r values thresholds is ignoring the motivation for this study to tease a subtle but apparently widespread signal from the euphotic zone, which is also supported by particulate and protein data here and which is found across two ocean sections.

The reviewer also comments on how: "Only the middle point is used in vector analysis". This is a misunderstanding. For each 5-depth point regression the resulting slopes are plotted centered around the data, which is the midpoint within the 5-points. We could easily switch the choice of depth to be any of the 5-depth points within the regression, but centering at its midpoint seemed most logical.

The reviewer's second main concern is regarding the lack of consideration of a mixing vector in the vector analysis. The reviewers incorrectly state that we "argue that only remineralization and scavenging govern the deep distribution of cobalt in the North Atlantic". On the contrary we state that "These scavenging signals co-occur with distinct water masses identified by OMPA analysis, implying that these scavenging processes are being integrated on decadal-to-century timescales of deepwater circulation processes within the ocean interior." While it is true that water mass mixing could alter the ratio of Co and P, mixing itself is not a process that removes or add Co or P from the dissolved phase (except in estuaries). Moreover, the ratios of the dCo and PO4 provided in the review produce an almost identical ratio of 32 and 31-35 (40, 70-80pM dCo; 1.25 and 2.25 PO4), so mixing these NADW and AABW water masses would not create a significant vector in Co:P space by their example. Our point here is that in these localized fine scale regression analyses vertical processes of uptake remineralization and

scavenging, whose signals are clearly integrated through horizontal advection, can explain the large variation and change in sign of the Co:P slopes. Finally, it is not clear that mixing would produce a single vector depending on the ratios being mixed, but instead could be in a number of directions. We did include a mixing vector in our original vector diagram for upwelling (Noble et al., 2008), but that may be a specific coastal scenario. As a result, we politely decline to include a mixing vector in the schematics at this time and instead focus on the chemical and biochemical processes that are capable of moving dissolved cobalt and phosphorus to the solid phase, rather than mixing processes that cannot.

The reviewer suggests using the water mass analysis in this study of end member distributions: "since the complete analysis of water-masses is available, as performed by Jenkins et al., 2015), the combination of end-members weighted by the proportion of each water-mass will help to determine if a located depth is a source (remineralization) or a sink (scavenging) for dCo in the deep sea." We point out that this water analysis has already been included and is shown in Figure 2 for the North and South Atlantic Zonal sections including the following text that relates to the reviewer's suggestion. Our manuscript states: "These scavenging signals co-occur with distinct water masses identified by OMPA analysis, implying that these scavenging processes are being integrated on decadal-to-century timescales of deepwater circulation processes within the ocean interior (Noble et al., submitted). Specifically, negative slopes water masses were found to be in the Denmark Straits Overflow Water/Antarctic Bottom Water/Iceland Scotland Overflow Water (DSOW/AABW/ISOW) and Classical Labrador Seawater (CLSW; Fig. 2) water masses both of which have long deepwater transit times (Jenkins et al., 2015)." We argue that the negative vectors we observe are the influence of slow scavenging processes accumulating on isopyncals during advection, and hence that the influence of scavenging and mixing are combined in the negative vectors we observe. The reviewers imply that mixing of Co is a conservative process (without any scavenging), but we respectively disagree with this argument from Dulaquais et al., 2014, and argue that the data shown here, and in a more substantive manuscript on Co scavenging (Hawco et al, Marine Chemistry in revision) provide evidence of mesopelagic scavenging. In particular, in the latter manuscript a comparison of dCo with C14 ages in DIC showed basin scale scavenging processes between the Atlantic and Pacific basins. Much of the reviewer's interest in scavenging is more pertinent to the Hawco et al. manuscript, while the present manuscript focuses on the dynamic variation in dCo:P relationships in the vertical dimension, which must include some discussion on the unusual negative slopes in the mesopelagic and deep ocean and how the shift from remineralization to scavenging can result in the transition in positive to negative slopes as shown in Figure 10.

The reviewer states that we are inconsistent with Moffett and Ho's Co and Mn microbial oxidation dataset which did not find measurable Mn oxidation at BATS. This is incorrect, Moffett and Ho found phytoplankton uptake dominates within the euphotic zone at 60m, which is consistent with our uptake/remineralization vector based on positive slopes at this depth. Moffett and Ho did not present uptake rates in the mesopelagic in that manuscript.

The reviewer suggests an alternate means for particulate Co-Mn correlations, "The strong Co-Mn correlation can be interpreted in another way than that actually proposed: it is indeed possible that the biogenic Co fraction is remineralized while the authigenic fraction (scavenged fraction) is not (i.e.; the one way trip of Co presented by authors)". We appreciate the reviewers considering and describing this scenario in detail. This interpretation is actually the phenomenon we are presenting and arguing for in every aspect they discuss (an increasing importance of scavenged/authigenic particulate for pCo and

pMn with depth resulting in the pCo:pMn relationship and the increased pCo:pP relationship). We are glad to hear they agree! We point out that it is this authigenic pCo formation process that in turn creates the negative dCo:dPO4 slopes in the dissolved phase we have been debating above. We have found that the connections between dissolved and particulate phases are fascinating yet can also be confusing and take time to fully consider given the mirroring/opposite effects the phases can have on each other and out community's limited experience in direct comparisons of dissolved and particulate data.

The reviewer states that the vector approach does not take into account differential remineralization. This is an interesting point, but upon discussion we think it is not true. The profile-based regression analysis does not include any a priori assumption about stoichiometry or remineralization efficiency. It is simply a measurement of dCo and dP change in response to each other in the vertical dimension. There were no Co* values reported in Hawco 2016 or Noble 2012, but it is here. I understand the hypothesis: an alternate explanation for vertical stoichiometric structure observed here is the preferential remineralization of P over Co creating the Co deficiency, as an alternative to scavenging. This is a useful alternate perspective that we have added to the discussion. Reflecting on it, there are several datasets that argue against differential remineralization being the major driver in dCo distributions: 1) the scavenged like profile is inconsistent with this (implying removal rather than slowed accumulation), 2) the correlation of pCo and pMn in the mesopelagic is consistent with biotic Mn oxidation rather than with slow Co remineralization (which would not create a correlation with Mn), and 3) the negative slopes of dCo:P are consistent with Co loss rather than slowed accumulation (which would be a positive slope still). To clarify, the prior and current observation of increased dCo in the OMZ plumes were not interpreted as a result of slowed / differential remineralization but reduced Mn and Co scavenging. This is an area that is ideal for future investigations that are ongoing by collaborators.

The reviewer suggests incorporating data from Dulaquais et al into the statistical analysis of this study and the table for aggregate analysis. We have incorporated the reviewer's published Co:P ratios into Table 1, and we thank her for this suggestion. As for incorporating other sections of data into the vector and profile-based regression analysis, that is beyond the scope of what we can conduct in this study. There is already a lot of data within this study and there were some intercalibration challenges for dCo at the western portion of the North Atlantic, as the reviewer is aware. Future efforts could apply this approach to other ocean sections of Co:P.

The reviewer comments on our choice of a Co* ratio from the pCo:pP North Atlantic ratio, and argues that it should instead use the lower Twining from phytoplankton species. This is a good comment and one we also wrestled with. We point out that Redfield et al's original studies used oceanic particulate carbon, nitrogen and phosphate ratios as well, and they have been interpreted as being reflective of the aggregate community, so there is precedent for working from the aggregate particulate phase. The cell specific numbers from Twining et al. are taxon-specific and span 140-3000 depending on the group (their Table 5). We were transparent about the challenge of selecting a single Co:P in the manuscript, yet we feel the cohesive structure in the visualization of Co* is compelling particularly in the mesopelagic demonstrating loss by scavenging, even if its interpretation is not fully understood as of yet. Based on this discussion we feel it is most representative to maintain use of the field values for Co:P in the revision. We point out that though that the choice of any the Co:P value will not affect the trends observed in this visualization, since any change in Co:P will be applied linearly to all Co* values.

The reviewer brings up the matter that dust is not considered much within this manuscript. We have corrected this by incorporating both references and lithogenic corrections of the particulate phase, presented in sections 3.2.2 and 3.3, and presented visually in Figure 11. In almost all cases, and all cases in the upper 400m, the lithogenic component is a minority constituent relative to the biological/authigenic component. Moreover, dust is discussed more extensively in the accompanying Noble et al., BG 2017 manuscript. We have added the following text to demonstrate that pCo is largely biogenic in the study region (although this does not negate the importance of the minority lithogenic phase for Co cycling, a subject beyond the scope of this manuscript):

P9-10: "These oceanic upper water column pCo values here appear to be largely associated with biogenic material: Twining et al. reported pCo in the upper 100m to be largely labile by their leaching methods (Twining et al., 2015), and analysis of the small size fraction from McLane pump samples found the lithogenic component to be a minority component in most samples, particularly in shallower samples (all samples shallower than 400m depth had a minority lithogenic contribution; see Section 3.3 below). Based on these observations and calculations, the assumption of pCo being dominated by biogenic and authigenic components appears reasonable for our stoichiometric discussion, although the observation of the presence of this minority component of lithogenic particulate cobalt in the deep ocean could contribute a slowly dissolving gradual source of dCo in the interior ocean that could be further studied, consistent with prior field and laboratory observations (Noble et al., 2017; Mackey et al., 2015)."

P17: "Lithogenic corrections included here for pCo and previously described for both pCo and pP (Ohnemus and Lam, 2015), show that these elements had minor lithogenic contributions in the North Atlantic particularly in the near surface and typically being a minority contribution at deeper depths, even in the heavily impacted North Atlantic region (Fig. 11a, c, e)."

Responses to specific comments:

Lithium batteries: see response to reviewer #1.

P5. Detection limit values were added, values below 0.02uM were not included from the GA03 dataset.

P7: Dulaquais comparison and reference added, although the value in the Dulaquais paper is 66 not 64.

P8: Regarding the measurement of labile cobalt: we disagree that with the reviewer's statement that it is also complexed and not equivalent to inorganic cobalt. Labile cobalt is indisputably in a much lower detection window that complexed cobalt, and is as a result likely Co(II) in redox state, and could be bound by inorganic or weak organic complexes, as we have documented previously (Saito et al., 2005). Our previous studies have shown one cyanobacterial species is able to take up the strongly bound cobalt, while our and other studies have clearly also shown there to be a inorganic cobalt transporter present in both cyanobacteria and eukaryotic algae as well (Saito et al., 2002, Sunda and Huntsman 1995 and others). We have changed the sentence to reflect this: "The presence of a substantial labile cobalt pool was consistent with the observed high cobalt usage, since labile cobalt is likely highly bioavailable relative to complexed cobalt, particularly for eukaryotic phytoplankton, due to its ability to be taken up through divalent cation transporters (Saito et al., 2002; Saito et al., 2005)."

On the recommendation that we use alternate Redfield ratios: this does not make sense in this context because we are converting laboratory cultures quota data that was grown at replete P conditions, hence standard Redfield is the correct approach.

Moffett and Ho: they only studied surface waters for Mn/Co oxidiation in that study.

Advection is not a process that removes Co from the dissolved phase relative to P (or P to particulate P), hence it is not included here.

dt is used because it reflects changes within a discrete water parcel over time, as is used in 1D models (Johnson et al., 1997; Weber et al., 2007).

Added labels to subfigure 4d, e, and f. Thank you.

Data from Dulaquais et al., 2014 was integrated into Table 1.

Figure 2. The scales are clearly labeled differently. South Atlantic Central Water (SACW) was not one of the water masses characterized in the Ruth et al. 2000, and the similar Saito et al. 2012 South Atlantic OMPA analyses. We have inquired about the possibility of reconstructing the South Atlantic OMPA analyses with similar parameters to the North Atlantic but if it happens it will not be available in time for this revision. The Ross Sea is included as Southern Hemisphere endmember.

Fig 3 caption fixed

Figure 4. Insert made into a panel g

Figure 4. depth was not added as a z parameter since this is illustrated in Fig 3. Figure b and e were integrated further into the discussion.

Figure 5 The reviewer states that "some bottom depths and the depths just above them are circled and it is not in accordance with the 5 point regression being used". The reviewer is pointing out that for the regression to be in the center of 5 vertical points, there should not be regressions at the base of the profiles.  For those profiles the bottom depths are actually off scale since this figure uses uniform axes, or the individual two datapoints below each circle are not visible in these small graphs. We have included a table of all regression analyses as supplemental data to allow the reader to examine this.

Figure 6: see discussion above about r values, and added r values in figure 3.

Figure 7: the reviewer asks for dCo data to be included. This data is already in Figure 5.

Figure 10: see discussion above on variability of a mixing vector.

Figure 12: while the r2 values are low, the slopes are negative, which is the intention of the presentation.

Figure 13: z coloring would obscure the current red black coloring scheme.

Figure 14. Isoclines were not added to avoid cluttering the figure.